# ENRICHING ONLINE KNOWLEDGE DISTILLATION WITH SPECIALIST ENSEMBLE

## ABSTRACT

Online Knowledge Distillation (KD) has an advantage over traditional KD works in that it removes the necessity for a pre-trained teacher. Indeed, an ensemble of small teachers has become typical guidance for a student's learning trajectory. Previous works emphasized diversity to create helpful ensemble knowledge and further argued that the size of diversity should be significant to prevent homogenization. This paper proposes a well-founded online KD framework with naturally derived specialists. In supervised learning, the parameters of a classifier are optimized by stochastic gradient descent based on a training dataset distribution. If the training dataset is shifted, the optimal point and corresponding parameters change accordingly, which is natural and explicit. We first introduce a label prior shift to induce evident diversity among the same teachers, which assigns a skewed label distribution to each teacher and simultaneously specializes them through importance sampling. Compared to previous works, our specialization achieves the highest level of diversity and maintains it throughout training. Second, we propose a new aggregation that uses post-compensation in specialist outputs and conventional model averaging. The aggregation empirically exhibits the advantage of ensemble calibration even if applied to previous diversity-eliciting methods. Finally, through extensive experiments, we demonstrate the efficacy of our framework on top-1 error rate, negative log-likelihood, and notably expected calibration error.

## 1 INTRODUCTION

Knowledge Distillation (KD) has achieved remarkable success in model compression literature (Heo et al., 2019; Park et al., 2019; Tung & Mori, 2019). KD traditionally employs a two-stage learning paradigm: training a large static model as a "teacher" and training a compact "student" model with the teacher's guidance. Online KD (He et al., 2016; Song & Chai, 2018; lan et al., 2018) emerged as a variant of KD, which simplifies the conventional two-stage pipeline by training all teachers and a student simultaneously. Previous works used a limited number of small teachers and treated them as auxiliary peers that help a student learn. Especially, ensembling these teachers has become a typical direction to make knowledge guidance for the student.

A core question in online KD is how to make teachers diverse for the ensemble. Breiman (1996) argues that traditional Bagging-style ensembles usually benefit from diverse and dissimilar models. Recent online KD studies (Chen et al., 2020; Li et al., 2020; Wu & Gong, 2021) support this claim and emphasize the importance of large diversity to prevent homogenization. In supervised learning, the parameters of a classifier are optimized by stochastic gradient descent based on a training data distribution. If the training dataset is shifted, the optimal point and corresponding parameters change accordingly, which is natural and explicit. That is, diversifying training data distribution sheds light on effectively generating diverse classifiers resorting to different features.

In this paper, we use *label prior shift*, where each teacher is assigned unique and non-uniform label distribution. This approach partially aligns with the specialization process in Mixture of Experts (MoE) literature, in which multiple experts with different problem spaces learn only the local landscape (Baldacchino et al., 2016). The most straightforward and prevalent approach to dealing with label imbalance is to operate on the shifted dataset itself (Japkowicz & Stephen, 2002; Chawla, 2009; Buda et al., 2018). However, online KD may have an inconvenient design that could necessitate sampling as much as the number of teachers because a typical framework has shared

layers in a multi-head architecture. As an alternative way, we consider adjusting a cross-entropy loss of each teacher rather than recursive sampling. Therefore, we efficiently estimate the loss functions using *importance sampling* drawn from the usual uniform label distribution instead of directly multi-sampling from the truly shifted distributions. Our specialization exhibits the highest level of diversity and maintains it throughout the training compared to prior works.

Furthermore, we propose a new ensemble strategy for aggregating specialist teacher outputs. From a perspective of Bayesian inference, it can be interpreted that the conditional distributions of specialists become likewise distorted when a classifier learns the label-imbalance training dataset. Therefore, we need to correct the distortion of conditional distributions before the aggregating process. We first use *PC-Softmax* (Hong et al., 2021) to post-compensate Softmax outputs. Post-compensation adapts the shifted label priors according to the true label prior by manually adjusting teacher logits. It relaxes the disparity in negative log-likelihoods (Ren et al., 2020) for the same label. As a result, PC-Softmax matches the uniform label distribution by modifying the teacher prediction trained by unique cross-entropy loss. Second, we apply a standard model averaging method (Li et al., 2021) to all the PC-Softmax outputs. We empirically show that our aggregation policy, denoted "specialist ensemble," improves ensemble calibration even when applied to previous diversity-eliciting methods.

Our main contributions are summarized as follows:

> (1) The proposed online knowledge distillation promotes diversifying teachers to be specialists through the label prior shift and importance sampling. As a result, our diversity is at the highest level over previous works and maintained throughout training
> (2) Our specialist ensemble, based on PC-Softmax and averaging those probabilities, is beneficial in ensemble calibration. Moreover, this advantage is valid even when applied to previous diversity-eliciting methods.
> (3) Through extensive experiments, we describe that a student distilled by our specialist ensemble outperforms previous works in top-1 error rate, negative log-likelihood, and notably expected calibration error.

## 2 RELATED WORK

**Label prior shift.** The label prior shift has been extensively discussed due to various degrees of imbalance in training (source) label prior $p_s(y)$ and test (target) label prior $p_t(y)$. Especially in most works closely related to ours, Post-Compensating (PC) strategy is typically chosen as the proper adjustment to estimate new conditional probability $p(y|x)$ approximated by $p_s(y)$ for given $p_t(y)$. When estimating a Softmax regression, Ren et al. (2020) corrects the model outputs by the amounts of each class, assuming the uniform target distribution during training time. Many strategies for matching two priors at test time were investigated by rebalancing a different form of multiplying $p_t(y)/p_s(y)$ to the output probability from a Bayesian perspective (Buda et al., 2018; Hong et al., 2021; Margineantu, 2000; Tian et al., 2020). Here, Hong et al. (2021) carefully reconstruct each conditional probability that should satisfy a condition $\sum_c p_t(y = c|x) = 1$. It is known as PC-Softmax. We use PC-Softmax of each teacher network to adapt entirely different label priors according to the student label prior.

**Ensemble learning.** Promoting diversity in traditional ensemble learning has been emphasized because the number of models, which acts as a factor of ensemble impact, becomes more crucial when they are gradually uncorrelated. (Breiman, 1996; Ghojogh & Crowley, 2019). Lakshminarayanan et al. (2017) used only different random initialization and weighted averaging on the same models. The models, as a result, can have similar error rates but converge to different local minima (Wen et al., 2020). Bringing in multiple models, however, requires prohibitively large computational resources, which frequently limits the ensemble's applicability. Thus, recent studies have found efficiencies in two approaches: sampling multiple learning trajectories with only a single model (Huang et al., 2017a; Laine & Aila, 2017; Tarvainen & Valpola, 2017) and building a new structure architecturally efficient (Wen et al., 2020; Li et al., 2021). Our ensemble overview can be aligned with the latter by modeling shared parameters and purposive heads to be diversified.

**Online knowledge distillation.** Online knowledge distillation works belong to two categories: network-based and peer-based. Network-based methods (Guo et al., 2020; Zhang et al., 2018) train

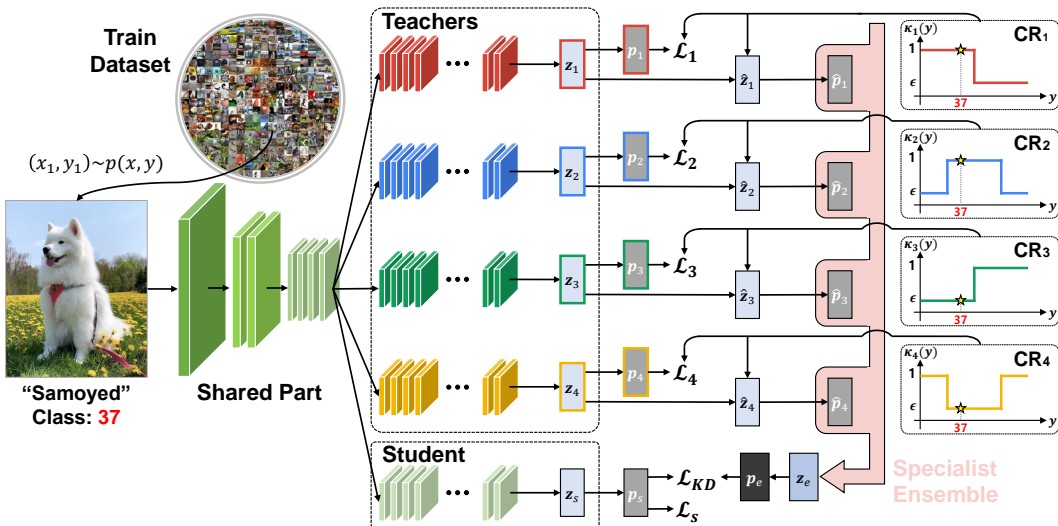

Figure 1: Overview of our online knowledge distillation framework with four teachers. Each teacher is assigned a different label prior by Class Reweighting (CR) function as described in Section 3.3. Each teacher loss and a student loss is defined in Section 3.4 and Section 3.6, respectively. For knowledge distillation, a specialist ensemble is obtained as described in Section 3.5.

separate networks with identical architecture, employing mutual learning paradigm; every network interacts and provides knowledge guidance to its cohorts. Sometimes, these allow independent data pre-processing per network. However, in peer-based methods (Song & Chai, 2018; Ian et al., 2018; Wu & Gong, 2021; Li et al., 2020), some parameters are shared among peer heads, which concurrently benefit from computational efficiencies and generalization during training. Earlier works exploit surrogate modules to enlarge varying features or introduce each head unfairly for peer diversity. However, following trained parameters, the modules may ignore some peers as useless, which prevents achieving the preferred diversity (Mullapudi et al., 2018). Chen et al. (2020); Kim et al. (2021) can be applied to both network- and peer-based approaches, but they require extra modules as well. Our method is peer-based and shares two similarities with some previous works: first, designating a "student" and "teachers" in advance of the training to equip one-way guidance (Chen et al., 2020; Li et al., 2020), and second, creating a dedicated dataset to train each peer (Feng et al., 2021). However, our specialty dataset differs from an arbitrary sampled subset (Feng et al., 2021) in that it is purposely class-skewed.

## 3 METHODOLOGY

As shown in Figure 1, our model consists of three parts: shared part, multiple teacher heads and the student head. That is, we parameterize it as $\Theta = \{\boldsymbol{\theta}_\phi\} \cup \{\boldsymbol{\theta}_t | t \in \mathbb{N}, t \leq T\} \cup \{\boldsymbol{\theta}_s\}$ where $T$ denotes the number of teachers. $\{\boldsymbol{\theta}_\phi\}$ represents the shared parameters; $\{\boldsymbol{\theta}_t\}$ and $\{\boldsymbol{\theta}_s\}$ represent the teacher and student head parameters, respectively. For simplicity, we will use the same notation for each teacher and student model including shared parameters, i.e., $\{\boldsymbol{\theta}_t\} = \{\boldsymbol{\theta}_\phi\} \cup \{\boldsymbol{\theta}_t\}$ and $\{\boldsymbol{\theta}_s\} = \{\boldsymbol{\theta}_\phi\} \cup \{\boldsymbol{\theta}_s\}$. Note that each $\{\boldsymbol{\theta}_t\}$ and $\{\boldsymbol{\theta}_s\}$ have the same dimension.

### 3.1 LABEL PRIOR SHIFT

Prior works (Buda et al., 2018; Hong et al., 2021; Margineantu, 2000; Tian et al., 2020) addressed the discrepancy between train and test class distributions due to the inherent difficulty in obtaining samples from certain classes and dealt with such class imbalance with post-scaling prior distributions. In our work, we adopt a similar method to manually shift the label prior distribution of teachers for the student to learn from diverse teachers.

We want the model output $p(y|x; \boldsymbol{\theta})$ to approximate a true posterior distribution $p(y|x)$, which is a conditional distribution of the labels $y$ given input samples $x$. From the perspective of Bayesian

inference, a true posterior distribution is defined as follows:

$$p(y|x) = \frac{p(x|y)p(y)}{p(x)}, \tag{1}$$

where $p(y)$ represents a label prior distribution. We assume $p(y)$ is a discrete uniform distribution $y \sim \mathbb{U}(1/K)$ where $K$ is the number of classes since the datasets we are using contain the uniform number of training samples for each class. In our setting, label prior shift refers to the label distribution distinction between the teachers and the student, i.e., $p_t(y) \neq p_s(y)$. While $p_s(y) = p(y)$, we purposely make class-imbalanced settings by shifting label distributions of teachers. Also, $p_t(y)$ differs from each teacher, i.e., $p_t(y) \neq p_{t'}(y)$, motivating each teacher to be a diverse discriminative classifier. We will discuss how to manipulate teacher label distributions in Section 3.3.

## 3.2 IMPORTANCE SAMPLING

Under our class-imbalance setting, naive Monte-Carlo sampling is unlikely to effectively approximate the target distribution. Therefore, we exploit *importance sampling*, which allows us to effectively approximate the target distribution only with the samples generated from a distribution we have.

We will denote the shifted label prior $p_t(y)$ as $q(y)$ only in this section to avoid confusion with $p(y)$.

**Theorem 1.** *Let $q(x, y)$ and $p(x, y)$ be joint probability distributions, and $h(x, y)$ be a differentiable function with respect to $x$ and $y$. Assuming $q(x|y) = p(x|y)$, we can estimate $\mu = \mathbb{E}_{(x,y) \sim q(x,y)}[h(x, y)]$ as follows:*

$$\mu = \mathbb{E}_{(x,y) \sim p(x,y)}\left[\frac{q(y)h(x, y)}{p(y)}\right] \approx \frac{1}{N}\sum_{i=1}^{N}\frac{q(y_i)h(x_i, y_i)}{p(y_i)}, \quad (x_i, y_i) \sim i.i.d. \ p(x, y). \tag{2}$$

During training, a sample $(x_i, y_i)$ is drawn from a joint distribution $p(x, y) = p(x|y)p(y)$. However, what we actually aim is to sample from the joint distribution with shifted prior $q(y)$, $q(x, y) = q(x|y)q(y)$. Theorem 1 shows we can effectively estimate the expectation of the function $h(x, y)$ where $(x, y) \sim q(x, y)$ with the samples drawn from $p(x, y)$.

Often the target label distribution $q(y)$ is intractable and we only know unnormalized $\tilde{q}(y) = Zq(y)$ with unknown normalization constant, $Z$. We make use of an unnormalized distribution, $\tilde{q}(y)$, by our design choice.

**Corollary 1.1.** *Let $\tilde{q}(x, y) = Zq(x, y)$ be an unnormalized distribution with unknown constant $Z > 0$. Then, the unnormalized importance sampling estimator of $\mu$ is as follows:*

$$\mu \approx \sum_{i=1}^{N}\frac{\kappa(y_i)h(x_i, y_i)}{\sum_{i=1}^{N}\kappa(y_i)}, \quad (x_i, y_i) \sim i.i.d. \ p(x, y), \tag{3}$$

*where $\kappa(y) = \tilde{q}(y)/p(y)$ is the unnormalized reweighting function of $y$.*

As shown in Corollary 1.1, we can obtain the estimator of the data loss $h(x, y)$ under label prior shift, once $\kappa(y)$ is given. In the following section, we will discuss how to formulate the function $\kappa(y)$.

## 3.3 CLASS REWEIGHTING FUNCTION

Our goal is to specialize each teacher in a specific subset of labels. To achieve this, each teacher is assigned "specialty" labels. We denote this "specialty" label set for $t$-th teacher as $\mathcal{Y}^t$. $\mathcal{Y}^t$s can overlap, but all labels are assigned at least once and then for the same number of times, i.e., $\mathcal{Y} = \mathcal{Y}^1 \cup \cdots \cup \mathcal{Y}^T$, where $\mathcal{Y}$ is the total label set. Although we sequentially allocate the label to teachers (see Appendix.B), clustering (Mullapudi et al., 2018) or human-crafted grouping (Krizhevsky, 2009) can be also considered.

Now we introduce a Class Reweighting (CR) function, $\kappa_t(y)$ to assign high weights to "specialty" labels $\mathcal{Y}^t$. This function indicates how much the $t$-th teacher considers the label $y$ against $p(y)$. As mentioned before, $p(y)$ is a uniform probability distribution. However, as $\tilde{p}_t(y)$ is an unnormalized (distribution), we consider it as a function of $y$.

$$\kappa_t(y) = \frac{\tilde{p}_t(y)}{p(y)} = \begin{cases} 1, & \text{if } y \in \mathcal{Y}^t \\ \epsilon, & \text{otherwise,} \end{cases} \quad 0 \leq \epsilon \leq 1. \tag{4}$$

We name $\epsilon$ as *exposure*. If exposure is 1, there is no label prior shift, so $\tilde{p}_t(y)$ is also a uniform distribution equal to the student label prior.

### 3.4 TEACHER LOSS

Neural networks typically produce class probabilities by using a "Softmax" output layer that converts the logit computed for each class into a probability by comparing it with the other logits. Let $z_t^i[k]$ denote the logit of class $k$ given an $i$-th input sample $(x_i, y_i)$ produced by the $t$-th teacher model. Then, the output conditional probability of Softmax layer is as follows:

$$p_t(y_i|x_i; \boldsymbol{\theta}_t) = \frac{\exp\left(z_t^i[y_i]\right)}{\sum_{k=1}^{K} \exp\left(z_t^i[k]\right)}, \quad t \leq T. \tag{5}$$

Using the unnormalized importance sampling estimator in Eq. 3, teacher loss for random samples $\{(x_i, y_i)\}_{i=1}^{N}$ where $N$ is the number of samples is defined as follows.

$$\mathcal{L}_t(\boldsymbol{\theta}_t) = \mathbb{E}_{(x,y)\sim p(x,y)}[-\kappa_t(y)\log(p_t(y|x; \boldsymbol{\theta}_t))] \approx -\sum_{i=1}^{N} \frac{\kappa_t(y_i)\log(p_t(y_i|x_i; \boldsymbol{\theta}_t))}{\sum_{i=1}^{N} \kappa_t(y_i)}, \ t \leq T. \tag{6}$$

### 3.5 SPECIALIST ENSEMBLE FOR KNOWLEDGE DISTILLATION

Following the model averaging paradigm (Li et al., 2021), we aggregate teachers' predictions to define a guide signal of knowledge distillation loss. The original predictions, however, have shortcomings.— $t$-th teacher conditional distribution $p_t(y|x; \boldsymbol{\theta}_t)$ is closely related to each prior $p_t(y)$; since supervision signals for minority classes are unlikely to occur, teachers may fail to introduce correct predictions on uniform $p(y)$. In order to adapt according to $p(y)$, we thus relax the minority classes' likelihood by manually adjusting logit values (Ren et al., 2020). We introduce further studies in Appendix.C.

**Adapting label prior.** We adjust teacher output logits to adapt shifted teacher priors $p_t(y)$ to uniform $p(y)$. Following discussions of Appendix.C, we employ PC-Softmax (Hong et al., 2021) to post-compensate teacher logits. Given the original teacher logits $z_t^i$ and CR function $\kappa_t(y)$ in Eq. 4, post-compensated logits (PC-Logits) and conditional probabilities of the $i$-th sample produced by the $t$-th teacher are defined as follows:

$$\hat{z}_t^i[y_i] = z_t^i[y_i] - \log\left(\frac{1}{\kappa_t(y_i)}\right); \quad \hat{p}_t(y_i|x_i; \boldsymbol{\theta}_t) = \frac{\exp\left(\hat{z}_t^i[y_i]\right)}{\sum_{k=1}^{K} \exp\left(\hat{z}_t^i[k]\right)}, \ t \leq T, \tag{7}$$

where $\kappa_t(y_i)$ is $\tilde{p}_t(y_i)/p(y_i)$ well defined by Section 3.3. Thus, if $\kappa_t(y_i)$ is 1, the corresponding class's logit is the same as the student's, but if $\kappa_t(y_i)$ is $\epsilon$, its logit is compensated as much as $-\log(1/\epsilon)$. Note that each head is trained with Eq. 5 and Eq. 7 is used only to compensate for the label distribution shift before making an ensemble prediction.

**Model averaging.** We now aggregate teacher predictions to form an ensemble prediction. Our fusion is based on an averaged classifier manner (French et al., 2018; Garipov et al., 2018), commonly used in statistical learning paradigm. Given conditional probabilities for an $i$-th sample, $\hat{p}_1(y_i|x_i; \boldsymbol{\theta}_1), \ldots, \hat{p}_T(y_i|x_i; \boldsymbol{\theta}_T)$, obtained from Eq 7, the aggregation is defined as follows:

$$p_e(y_i|x_i; \boldsymbol{\theta}_1, ..., \boldsymbol{\theta}_T) = \sum_{t=1}^{T} \hat{p}_t(y_i|x_i; \boldsymbol{\theta}_t)p(\boldsymbol{\theta}_t), \quad p(\boldsymbol{\theta}_t) = \mathbb{U}(1/T). \tag{8}$$

Each $\boldsymbol{\theta}_t$ is uniformly chosen. We also employ logarithm for the aggregated probabilities (Stanton et al., 2021) to derive a class ensemble logit $z_e^i[y_i] = \log(p_e(y_i|x_i; \boldsymbol{\theta}_1, ..., \boldsymbol{\theta}_T))$. In Section 4.3, we will compare this method to a convention of aggregating naive logits and show that our method has advantages in calibration.

### 3.6 STUDENT LOSS AND DISTILLATION STEPS

Given an ensemble logit and a student logit, here we define the cross-entropy and a knowledge distillation (Hinton et al., 2015) loss to update the student parameters $\{\boldsymbol{\theta}_s\}$. A temperature $\tau$ is used

---

**Algorithm 1** Student Distilling Steps

---

**Input:** Training set $\mathcal{D}$; $(T+1)$-head model parameterized $\Theta$; mini-batch size $N$; learning rate $\eta$
**Output:** A student model parameterized $\boldsymbol{\theta}_s^{converged}$
 1: Randomly initialize $\Theta$
 2: Set CR functions $\kappa_t(y)$ with a choice of $\epsilon$ to define each $\mathcal{L}_t(\boldsymbol{\theta}_t)$
 3: **while** $\boldsymbol{\theta}_s$ not converged **do**
 4:    $e \leftarrow e + 1$                                   ▷ Update epoch to adjust a ramp-up $\lambda(e)$ of Eq.13
 5:    **for** sample a mini-batch $\{(x_i, y_i)\}_{i=1}^N \sim \mathcal{D}$ **do**
 6:       Compute the entire $\mathcal{L}_t(\boldsymbol{\theta}_t)$ and $\mathcal{L}_s(\boldsymbol{\theta}_s)$ concurrently         ▷ Use Eq.6 and Eq.13
 7:       $\boldsymbol{\theta}_t \leftarrow \boldsymbol{\theta}_t - \eta\nabla_{\boldsymbol{\theta}_t}\mathcal{L}_t(\boldsymbol{\theta}_t)$
 8:       $\boldsymbol{\theta}_s \leftarrow \boldsymbol{\theta}_s - \eta\nabla_{\boldsymbol{\theta}_s}\mathcal{L}_s(\boldsymbol{\theta}_s)$
 9:    **end for**
10: **end while**

---

to soften probability distribution over classes. Same as Section 3.4, student loss is also defined by the same random samples.

$$p_s(y_i|x_i;\boldsymbol{\theta}_s) = \frac{\exp\left(z_s^i[y_i]/\tau\right)}{\sum_{k=1}^K \exp\left(z_s^i[k]/\tau\right)}; \quad p_e(y_i|x_i;\boldsymbol{\theta}_1,...,\boldsymbol{\theta}_T) = \frac{\exp\left(z_e^i[y_i]/\tau\right)}{\sum_{k=1}^K \exp\left(z_e^i[k]/\tau\right)}, \quad (9)$$

For normal cross entropy loss, $\mathcal{L}_{CE}$, temperature $\tau$ is set to 1. Knowledge distillation loss, $\mathcal{L}_{KD}$, is KL-divergence between the student and teacher ensemble posterior distributions.

$$\mathcal{L}_{CE}(\boldsymbol{\theta}_s) = \mathbb{E}_{(x,y)\sim p(x,y)}[-\log(p_s(y|x;\boldsymbol{\theta}_s))] \approx -\sum_{i=1}^N \log(p_s(y_i|x_i;\boldsymbol{\theta}_s)). \quad (10)$$

$$\mathcal{L}_{KD}(\boldsymbol{\theta}_s) = \mathbb{E}_{(x,y)\sim p(x,y)}[KL(\mathbf{p}_e||\mathbf{p}_s)] \quad (11)$$

$$\approx \sum_{i=1}^N \sum_{k=1}^K p_e(k_i|x_i;\boldsymbol{\theta}_1,...,\boldsymbol{\theta}_T) \log\left(\frac{p_e(k_i|x_i;\boldsymbol{\theta}_1,...,\boldsymbol{\theta}_T)}{p_s(k_i|x_i;\boldsymbol{\theta}_s)}\right). \quad (12)$$

The final student loss is a weighted sum of $\mathcal{L}_{CE}$ and $\mathcal{L}_{KD}$. We adjust $\lambda$ using a Gaussian ramp-up curve, which is $\lambda(e) = \exp(-5(1 - e/\alpha)^2)$, where $e$ is an epoch and $\alpha$ is the ramp-up period (Laine & Aila, 2017).

$$\mathcal{L}_s(\boldsymbol{\theta}_s) = \mathcal{L}_{CE}(\boldsymbol{\theta}_s) + \tau^2\lambda(e)\mathcal{L}_{KD}(\boldsymbol{\theta}_s). \quad (13)$$

Alg. 1 introduces student distilling steps. All parameters $\Theta$ are updated during training, and only the student, $\{\boldsymbol{\theta}_s\}$, is used at test time. Thus, our framework does not induce additional test-time costs.

## 4 EXPERIMENTS

In this section, we conduct three experiments to assess the efficacy of the proposed method. First, we evaluate how well our student model is generalized in an image classification compared to previous methods with three measurements (Stanton et al., 2021): Top-1 error rate (ERR), expected calibration error (ECE), and negative log-likelihood (NLL). Second, we perform an ablation study on the exposure $\epsilon$ of the proposed CR function $\kappa(y)$ and the number of teachers $T$. For the analysis, we include two metrics (Stanton et al., 2021) to measure a student's fidelity on an ensemble's outputs: averaged top-1 agreement and averaged KL-divergence. We further show diversity change following the variation of ramp-up period $\alpha$ in Appendix F.2. Finally, we empirically analyze why our student model has become calibrated. The evaluation settings are thoroughly summarized in Appendix E.1.

### 4.1 IMAGE CLASSIFICATION PERFORMANCE

We compare our method to extensive online KD methods: CLILR (Song & Chai, 2018), ONE (lan et al., 2018), FFL-S (Kim et al., 2021) and OKDDip (Chen et al., 2020) for CIFAR, as well as DML (Zhang et al., 2018), KDCL (Guo et al., 2020) and PCL (Wu & Gong, 2021) for ImageNet. Denoted "Vanilla" is to train a target model from scratch without knowledge distillation loss. While CLILR, ONE, and FFL-S select the first network as a student after the whole training procedure,

Table 1: The generalization comparison with previous peer-based methods on the student model. ERR and ECE use a percentage (%), and NLL is a loss value. Thus, the lower it is, the better. The numbers are the test results of three random experiments and filled in the mean(±std). The best result within each type is indicated in bold.

| Dataset | Method | ResNet-32 | | | ResNet-110 | | | DenseNet-40-12 | | | EfficientNetB0 | | | MobileNetV2 | | |
|---|---|---|---|---|---|---|---|---|---|---|---|---|---|---|---|---|
| | | ERR | ECE | NLL | ERR | ECE | NLL | ERR | ECE | NLL | ERR | ECE | NLL | ERR | ECE | NLL |
| CIFAR-10 | Vanilla | 6.30 (±0.15) | 3.98 (±0.20) | 0.30 (±0.14) | 5.35 (±0.27) | 2.98 (±0.16) | 0.36 (±0.19) | 6.90 (±0.09) | 3.47 (±0.04) | 0.29 (±0.13) | 8.68 (±0.19) | 4.65 (±0.06) | 0.27 (±0.00) | 11.34 (±0.14) | 3.14 (±0.17) | 0.34 (±0.00) |
| | CLILR | 5.65 (±0.18) | 3.38 (±0.05) | 0.23 (±0.00) | 4.52 (±0.10) | 2.55 (±0.15) | 0.18 (±0.01) | 7.05 (±0.06) | 2.96 (±0.22) | 0.24 (±0.01) | 7.47 (±0.12) | 3.60 (±0.12) | 0.28 (±0.01) | 11.76 (±0.27) | 2.56 (±0.24) | 0.35 (±0.00) |
| | ONE | 5.73 (±0.13) | 3.35 (±0.13) | 0.23 (±0.00) | 4.75 (±0.13) | 2.81 (±0.15) | 0.19 (±0.00) | 6.84 (±0.31) | 3.16 (±0.24) | 0.25 (±0.00) | 7.47 (±0.29) | 3.70 (±0.17) | 0.28 (±0.02) | 11.94 (±0.33) | 2.19 (±0.17) | 0.36 (±0.00) |
| | FFL-S | 6.04 (±0.19) | 4.40 (±0.19) | 0.29 (±0.01) | 4.55 (±0.05) | 3.27 (±0.09) | 0.23 (±0.00) | 6.90 (±0.08) | 4.36 (±0.15) | 0.32 (±0.01) | 7.43 (±0.17) | 3.88 (±0.21) | 0.28 (±0.02) | 11.40 (±0.17) | 2.45 (±0.21) | 0.34 (±0.00) |
| | OKDDip | 5.76 (±0.06) | 3.39 (±0.18) | 0.23 (±0.01) | 4.68 (±0.08) | 2.62 (±0.07) | 0.18 (±0.00) | 6.87 (±0.16) | 3.04 (±0.12) | 0.24 (±0.00) | 7.53 (±0.17) | 3.69 (±0.22) | 0.29 (±0.01) | 11.49 (±0.08) | 2.40 (±0.25) | 0.35 (±0.00) |
| | PCL | 6.12 (±0.14) | 3.76 (±0.41) | 0.25 (±0.02) | 4.77 (±0.25) | 3.42 (±0.21) | 0.23 (±0.00) | 6.84 (±0.14) | 3.61 (±0.04) | 0.25 (±0.00) | 7.12 (±0.09) | 3.81 (±0.13) | 0.25 (±0.00) | 11.35 (±0.26) | 3.08 (±0.39) | 0.35 (±0.00) |
| | Ours | **5.61** (±0.05) | **3.14** (±0.27) | **0.23** (±0.01) | **4.49** (±0.12) | **2.29** (±0.15) | **0.17** (±0.00) | **6.78** (±0.16) | **2.82** (±0.25) | **0.24** (±0.01) | **7.08** (±0.12) | **2.55** (±0.18) | **0.24** (±0.00) | **11.27** (±0.13) | **2.00** (±0.05) | **0.34** (±0.00) |
| CIFAR-100 | Vanilla | 28.70 (±0.21) | 13.04 (±0.28) | 1.22 (±0.33) | 24.34 (±0.22) | 12.38 (±0.37) | 1.27 (±0.11) | 28.58 (±0.07) | 9.06 (±0.27) | 1.25 (±0.18) | 29.47 (±0.29) | 13.22 (±0.28) | 1.17 (±0.02) | 35.04 (±0.86) | 5.82 (±0.15) | 1.24 (±0.02) |
| | CLILR | 26.45 (±0.16) | 6.81 (±0.29) | 0.99 (±0.00) | 21.49 (±0.18) | 9.36 (±0.57) | 0.88 (±0.02) | 28.51 (±0.13) | 5.99 (±0.72) | 1.04 (±0.02) | 27.70 (±0.35) | 10.48 (±0.14) | 1.15 (±0.06) | 33.21 (±0.41) | 4.37 (±0.53) | 1.18 (±0.01) |
| | ONE | 26.19 (±0.20) | 5.41 (±0.32) | 0.94 (±0.00) | 21.58 (±0.29) | 7.95 (±0.22) | 0.86 (±0.00) | 29.10 (±0.46) | 6.39 (±0.41) | 1.05 (±0.02) | 27.72 (±0.36) | 9.79 (±0.22) | 1.15 (±0.02) | 33.03 (±0.16) | 4.21 (±0.58) | 1.17 (±0.01) |
| | FFL-S | 26.19 (±0.07) | 10.60 (±0.28) | 1.05 (±0.01) | 22.15 (±0.31) | 9.58 (±0.90) | 0.92 (±0.04) | 28.95 (±0.18) | 10.35 (±0.20) | 1.13 (±0.01) | 27.61 (±0.11) | 10.81 (±0.34) | 1.15 (±0.00) | 33.52 (±0.57) | 6.04 (±0.53) | 1.21 (±0.02) |
| | OKDDip | 26.08 (±0.41) | 7.78 (±0.53) | 0.97 (±0.01) | 21.34 (±0.46) | 10.51 (±0.25) | 0.92 (±0.02) | 29.25 (±0.38) | 5.45 (±0.11) | 1.04 (±0.00) | 28.17 (±0.29) | 10.73 (±0.49) | 1.16 (±0.02) | 32.56 (±0.21) | 3.42 (±0.96) | 1.15 (±0.01) |
| | PCL | 26.78 (±0.26) | 10.56 (±0.85) | 1.09 (±0.03) | 21.02 (±0.21) | 10.81 (±0.31) | 0.92 (±0.01) | 29.10 (±0.43) | 10.85 (±0.27) | 1.15 (±0.01) | 27.59 (±0.75) | 11.91 (±0.62) | 1.15 (±0.04) | 34.94 (±0.47) | 11.46 (±0.71) | 1.37 (±0.02) |
| | Ours | **25.68** (±0.19) | **5.30** (±0.50) | **0.93** (±0.01) | **20.94** (±0.11) | **6.68** (±0.83) | **0.80** (±0.02) | **28.33** (±0.31) | **5.21** (±0.01) | **1.02** (±0.01) | **27.56** (±0.04) | **9.75** (±0.21) | **1.15** (±0.00) | **32.41** (±0.27) | **3.01** (±0.48) | **1.13** (±0.01) |

Table 2: Top-1 ERR (%) comparison with previous methods on ImageNet validation set. The results of ResNet-18 and ResNet-34 are each reported from Wu & Gong (2021) and Chen et al. (2020); Note also ours has $T = 2$ and $T = 3$ on each model for a fair comparison. We filled in mean(±std) through three random experiments on our validation results.

| Model | Vanilla | DML | CLILR | ONE | FFL-S | OKDDip | KDCL | PCL | Ours |
|---|---|---|---|---|---|---|---|---|---|
| ResNet-18 | 30.49(±0.14) | 30.18(±0.08) | 29.96(±0.05) | 29.82(±0.23) | 31.15(±0.07) | 30.07(±0.06) | 30.40(±0.05) | 29.58(±0.13) | **29.53(±0.03)** |
| ResNet-34 | 26.76 | 26.03 | 26.06 | 25.92 | 25.53 | 25.60 | - | - | **25.45(±0.09)** |

OKDDip and ours designate a student and other peers (teachers) in advance. Thus, OKDDip and our approach use one less peer than the others. All methods have total four heads. We evaluate these methods on various deep neural networks (DNNs), i.e. ResNet-32 and ResNet-110 (He et al., 2016)[1], DenseNet-40-12 (Huang et al., 2017b), EfficientNetB0 (Tan & Le, 2019), MobileNetV2 (Sandler et al., 2018) on CIFAR as well as ResNet-18 and ResNet-34 (He et al., 2016) for ImageNet. For building strategies of the peer-based architectures, we describe details in Appendix E.2.

**Results on CIFAR datasets.** Table 1 demonstrates that our method consistently outperforms previous methods to generalize a student. For all DNN models, our ERR and NLL are marginally better. However, our proposed method produces a significantly calibrated student in ECE than in previous works and Vanilla; the gains improve as the class size increases. Section 4.3 will discuss ECE further. We provide an additional comparison with network-based methods in Appendix F.1.

**Results on ImageNet datasets.** In comparison to all the previous methods, Table 2 shows Top-1 ERR. Our proposed method improves 0.96% and 1.29% ERR against Vanilla for each ResNet-18 and ResNet-34 and still achieves competitive superior among previous methods.

## 4.2 ABLATION STUDY

We examine the effectiveness of exposure variation ($\epsilon$) and the number of teacher heads ($T$). As shown in Figure 2, we analyze results for teacher diversity, student generalization, and student fidelity on the ensemble posterior distribution. We utilize averaged pairwise Jensen-Shannon divergence (Appendix D) to measure diversity between given two distributions. We use ResNet-110 trained on CIFAR-100 when $T = [2, 3, 4]$ and $\epsilon = [0.1, 0.3, 0.5, 0.8, 1.0]$; Models with more than five peers are not practical due to computational efficiency and saturation performance (Stanton et al., 2021). The range of $\epsilon$ is for our grid search. PC-Softmax is equally processed to evaluate the ensemble outcomes since $p(y)$ is the same in the training and testing data distributions.

---

[1]The model architecture of CIFAR is shallower than the plain version of ImageNet (He et al., 2016)

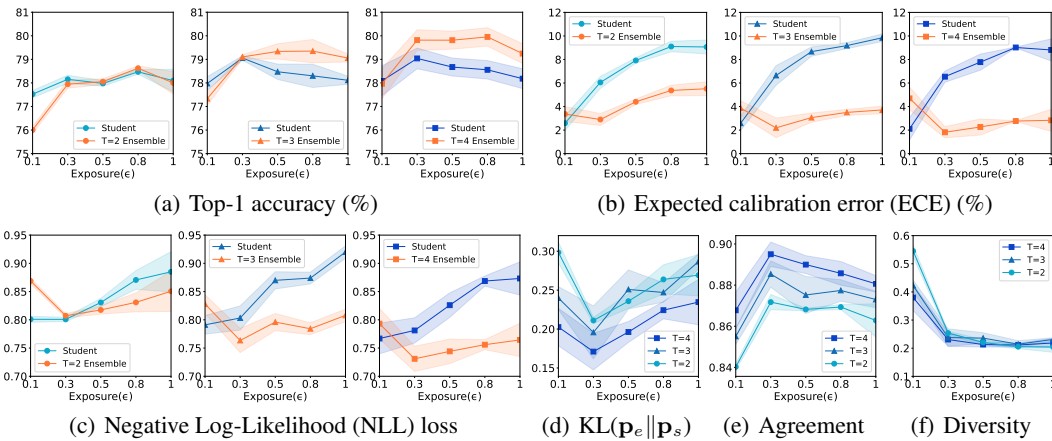

(a) Top-1 accuracy (%)  (b) Expected calibration error (ECE) (%)

(c) Negative Log-Likelihood (NLL) loss  (d) KL($\mathbf{p}_e \| \mathbf{p}_s$)  (e) Agreement  (f) Diversity

Figure 2: **Ensemble and student generalization**: top-1 accuracy, expected calibration error (ECE), and negative log-likelihood (NLL) loss. **Fidelity between ensemble and student conditional distribution**: averaged KL-divergence and averaged ensemble-student top-1 agreement. **Diversity**: averaged Jensen-Shannon divergence between the posterior distributions of each pair of teachers. The shaded region represents the mean($\pm$std) for three experiments with varying $\epsilon$ and $T$ in test time.

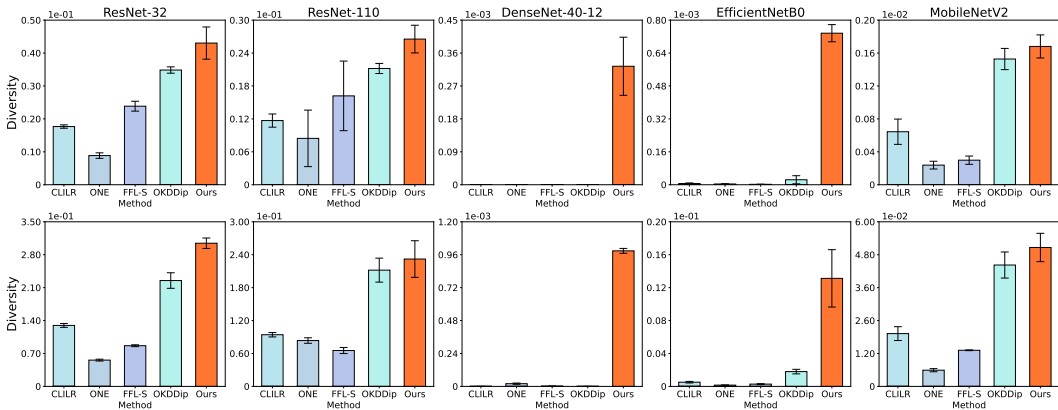

Figure 3: Diversity comparison in various deep neural networks on CIFAR-10 (**up**) and CIFAR-100 (**down**) with previous methods. For fair comparisons, we use Softmax to normalize teacher logits of the previous methods and PC-Softmax on our teacher logits. Each measure is obtained when $T = 3$ and the student is the best performer in validation time.

**Exposure variation.** As discussed in Section 3, $\epsilon$ determines teacher diversity. The diversity is exceptionally higher when $\epsilon$ is 0.1 as shown in Figure 2(f). At this point, the ensemble performs worse than a student, implying that the ensemble usually fails to discover the hidden knowledge in data. As a result, the student has high disagreements against the ensemble; this implies that the student may experience significant confusion during distillation. Diversity is lower than 0.1 in the other ranges, decreasing in small amounts from 0.3. The value $\epsilon$, 0.3, in particular, is quite encouraging. An outstanding generalized ensemble presents the potential to merge diverse teachers. As shown in Figure 2(d) and 2(e), the fidelity is also superior. The disparity in generalization is thus noticeably small in Figure 2(a) to 2(c). Furthermore, as shown in Figure 3, our diversity size by chosen $\epsilon$ in various DNNs presents consistently high compared to earlier methods. In Appendix F.3, we further visualize how teachers' confidence varies while predicting specific $k$-class samples.

**The number of teacher heads.** The fidelity typically improves as $T$ increases as shown in Figure 2(d) and 2(e). One possible explanation is that increasing the number of ensemble components smooths the logits of unlikely classes, making the distribution easier for the student to match. This phenomenon may provide insight into how to improve overall fidelity. The student thus benefits from top-1 accuracy and NLL loss, which improves ECE marginally, as shown in Figure 2(a) to 2(c). However, student generalization becomes increasingly saturated (Stanton et al., 2021) as $T$ increases. Meanwhile, diversity falls slightly because label repetition can render class coverage redundant.

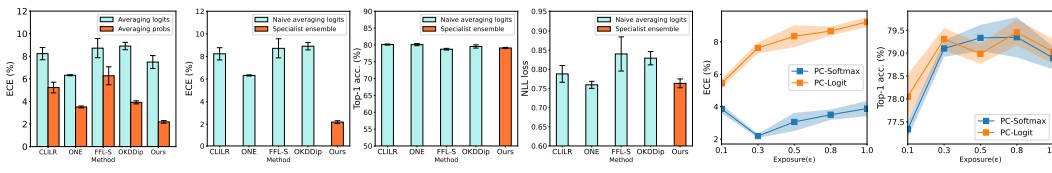

(a) Aggregation  (b) Generalizability in benchmark ensembles  (c) Choice of PC in our aggregation

Figure 4: Performance comparisons in ensembles using logits or probabilities (probs). All ensembles over the benchmarks are obtained when each student performs the best on accuracy at validation time. In particular, **(a)** probs are based on PC-Softmax for ours and Softmax for others. **(c)** The shaded region represents the mean($\pm$std), calculated from three trials with varying $\epsilon$

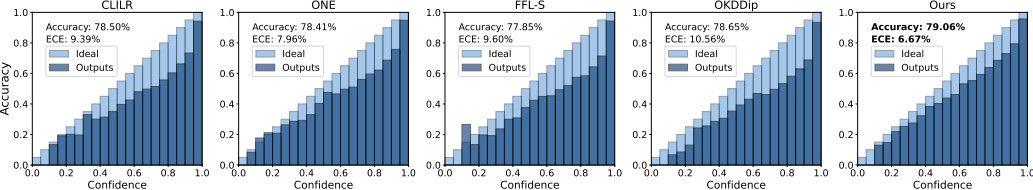

Figure 5: Reliability diagrams of student model for ResNet-110 on CIFAR-100. The confidence intervals are divided into 20-bins to visualize outcomes. Each output bars and assessments, such as accuracy and ECE, are a mean of three test experiments.

## 4.3 ON CALIBRATION OF STUDENT MODEL

This section empirically explains why our ensemble usually leads to better student calibration than previous methods. The calibration considers the problem of predicting probability estimates representative of the true correctness likelihood (Guo et al., 2017). KD can regard a type of learned label smoothing regularization (Yuan et al., 2020). The label smoothing can also calibrate a network, minimizing the miscalibration rate, i.e., ECE (Müller et al., 2019). Accepting such a KD effect, we conjecture two factors that our ensemble holds to transfer crucial scaling constraints for the student confidence: combining probabilities and diversity.

**Combining probabilities.** As shown in Figure 4(b), the posterior ensemble distribution with teacher probabilities rather than teacher logits significantly improves ECE and marginal gains with top-1 accuracy and NLL; we further provide ensembled confidence among them in Appendix F.4. Even after replacing the existing ensemble in previous methods with the probability-based and altering our ensemble to the logit-based, using probability still outperforms in ECE, as shown in 4(a). Moreover, as shown in Figure 4(c), PC-Softmax outperforms PC-Logit in ECE, exhibiting comparable accuracy in varying $\epsilon$. Through three case studies, we hypothesize that a probability-based ensemble effectively regularizes student confidence by KD guidance.

**Diversity.** As shown in Figure 3, our diversity shows higher and more model-agnostic than previous works; previous works have trouble deriving diversity on DenseNet-40-12 and EfficientNetB0. It implies that when the number of teachers is constrained, using extra losses may fail to induce a helpful diversity. Apart from the size and robustness of our diversity, acceptable fidelity demonstrates that the diversity is implicitly fine for a student to accommodate different signals, as shown in Figure 2(d) and 2(e). Therefore, we know that a merged knowledge is made of our useful diverse teachers to a direction suitable to a student, and it exhibits generalized ensemble performance as shown in Figure 2(a) to 2(c). The student, as a result, can learn generalized potential knowledge well.

## 5 CONCLUSION

We propose enriching online knowledge distillation with the specialist ensemble. Proposed CR functions are equipped to model label prior shifts for large diversity among teachers throughout training. Averaging diverse teacher probabilities provides a significant advantage in ensemble calibration. This paper confirms KD with our ensemble enlarges student generalization: marginal improved ERR and NLL with notable ECE. Figure 5 shows our student becomes a more predictable classifier than previous methods through reliability diagrams. For further discussions, the limitations and societal impacts are described in Appendices G and H.

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

# A PROOFS

## A.1 PROOF OF THEOREM 1

*Proof.* For discrete random variables $x$ and $y$, each joint probability mass function $p(x, y)$ and $q(x, y)$ can be expressed as $p(x, y) = p(x|y)p(y)$ and $q(x, y) = q(x|y)q(y)$ by *product rule*. We assume that $p(x|y) = q(x|y)$ and only label priors are different $p(y) \neq q(y)$. Also, let $h(x, y)$ be a differentiable function with respect to $x$ and $y$.

$$
\begin{aligned}
\mathbb{E}_{(x,y)\sim q(x,y)}[h(x,y)] &= \sum_x \sum_y q(x,y)h(x,y) \\
&= \sum_y q(y) \sum_x p(x|y)h(x,y) \\
&= \sum_y p(y)\frac{q(y)}{p(y)} \sum_x p(x|y)h(x,y) \\
&= \sum_y p(y)\frac{q(y)}{p(y)} (\mathbb{E}_{x\sim p(x|y)}[h(x,y)]) \\
&= \mathbb{E}_{y\sim p(y)}[\frac{q(y)}{p(y)}\mathbb{E}_{x\sim p(x|y)}[h(x,y)]]
\end{aligned}
\tag{14}
$$

By the associative law of multiplication and $y$ is a constant with respect to $x$, then we have

$$
\mathbb{E}_{y\sim p(y)}[\frac{q(y)}{p(y)}\mathbb{E}_{x\sim p(x|y)}[h(x,y)]] = \mathbb{E}_{y\sim p(y),x\sim p(x|y)}[\frac{q(y)h(x,y)}{p(y)}]
\tag{15}
$$

$$
= \mathbb{E}_{(x,y)\sim p(x,y)}[\frac{q(y)h(x,y)}{p(y)}].
\tag{16}
$$

One can wonder about the case $p(y = y_i) = 0$ for some $i$, so the denominator becomes zero. In our setting, $p(y)$ is a uniform distribution, thus, all the probabilities are strictly positive. $\qquad\square$

## A.2 PROOF OF COROLLARY 1.1

*Proof.* Let $\tilde{q}(x, y) = Zq(x, y)$ be an unnormalized distribution where $Z > 0$ is an unknown constant, then

$$
\sum_x \sum_y q(x,y) = \sum_x \sum_y p(x,y) = 1,
$$

$$
\sum_x \sum_y \tilde{q}(x,y) = Z.
$$

For every sample $(x_i, y_i) \sim$ i.i.d. $p(x, y)$, then $\hat{\mu} = \frac{1}{N}\sum_{i=1}^{N} h(x_i, y_i)$ is a basic Monte-Carlo estimator of $\mu = \mathbb{E}_{(x,y)\sim p(x,y)}[h(x,y)] = \sum_x \sum_y p(x,y)h(x,y)$. From Eq. 16,

$$
\begin{aligned}
\mu = \mathbb{E}_{(x,y)\sim p(x,y)}[\frac{q(y)h(x,y)}{p(y)}] &= \mathbb{E}_{(x,y)\sim p(x,y)}[\frac{\tilde{q}(y)h(x,y)}{Zp(y)}] \\
&= \sum_x \sum_y \frac{\tilde{q}(y)h(x,y)}{Zp(y)}p(x,y) \\
&= \frac{1}{Z}\sum_x \sum_y \frac{\tilde{q}(y)h(x,y)}{p(y)}p(x,y) \\
&= \frac{\sum_y [p(y)\frac{\tilde{q}(y)}{p(y)}[\sum_x p(x|y)h(x,y)]]}{\sum_y [\tilde{q}(y)[\sum_x \tilde{q}(x|y)]]} \\
&= \frac{\sum_y [p(y)\frac{\tilde{q}(y)}{p(y)}[\sum_x p(x|y)h(x,y)]]}{\sum_y [p(y)\frac{\tilde{q}(y)}{p(y)}[\sum_x \tilde{q}(x|y)]]}
\end{aligned}
\tag{17}
$$

Let $\kappa(y) = \frac{\tilde{q}(y)}{p(y)}$ as the unnormalized class reweighting (CR) function of $y$ and $\tilde{q}(x|y) = p(x|y)$, then we have

$$
\begin{aligned}
\frac{\sum_y [p(y) \frac{\tilde{q}(y)}{p(y)} [\sum_x p(x|y) h(x,y)]]}{\sum_y [p(y) \frac{\tilde{q}(y)}{p(y)} [\sum_x \tilde{q}(x|y)]]} &= \frac{\sum_y [p(y) \kappa(y) [\sum_x p(x|y) h(x,y)]]}{\sum_y [p(y) \kappa(y) [\sum_x p(x|y)]]} \\
&= \frac{\mathbb{E}_{y \sim p(y)} [\kappa(y) [\mathbb{E}_{x \sim p(x|y)} h(x,y)]]}{\mathbb{E}_{y \sim p(y), x \sim p(x|y)} [\kappa(y)]} \\
&= \frac{\mathbb{E}_{y \sim p(y), x \sim p(x|y)} [\kappa(y) h(x,y)]}{\mathbb{E}_{y \sim p(y), x \sim p(x|y)} [\kappa(y)]} \\
&= \frac{\mathbb{E}_{(x,y) \sim p(x,y)} [\kappa(y) h(x,y)]}{\mathbb{E}_{(x,y) \sim p(x,y)} [\kappa(y)]},
\end{aligned}
\tag{18}
$$

Using Monte-Carlo estimation, we can estimate the above expectations:

$$
\frac{\mathbb{E}_{(x,y) \sim p(x,y)} [\kappa(y) h(x,y)]}{\mathbb{E}_{(x,y) \sim p(x,y)} [\kappa(y)]} \approx \frac{\frac{1}{N} \sum_{i=1}^{N} \kappa(y_i) h(x_i, y_i)}{\frac{1}{N} \sum_{i=1}^{N} \kappa(y_i)}, \quad (x_i, y_i) \sim \text{i.i.d. } p(x,y)
\tag{19}
$$

$$
= \frac{\sum_{i=1}^{N} \kappa(y_i) h(x_i, y_i)}{\sum_{i=1}^{N} \kappa(y_i)}, \quad (x_i, y_i) \sim \text{i.i.d. } p(x,y).
\tag{20}
$$

$\square$

## B    FORMULATION OF SPECIALITY LABELS

This section describes how we create "specialty" labels $\mathcal{Y}^t$ of each $t$-th teacher (we already know $t \in \mathbb{N}, t \leq T$ where $T$ denotes the number of teachers). We allow the overlap between $\mathcal{Y}^t$s as mentioned in the main paper. We introduce a parameter $\gamma \in [0, 1]$ to control the ratio of the classes that a teacher focuses on. Let us denote the first class of $\mathcal{Y}^t$ as follows:

$$
c_0^t = \frac{K}{T}(t-1) + 1,
\tag{21}
$$

where $K$ is the total number of classes. Then, we can define the specialty set as:

$$
\mathcal{Y}^t = \{k \mid c_0^t \leq k \leq c_0^t + \gamma K - 1\}, \quad \text{if } c_0^t + \gamma K - 1 \leq K,
\tag{22}
$$

Sometimes $k$ can go beyond the last index of the whole class set. In that case, we define the specialty labels $\mathcal{Y}^t$ as follows:

$$
\mathcal{Y}^t = \{k \mid c_0^t \leq k \leq K\} \cup \{k \mid 1 \leq k \leq c_0^t + (\gamma - 1)K - 1\}, \quad \text{otherwise.}
\tag{23}
$$

In this paper, $\gamma = 0.5$ is used to let specialty labels overlap at least once. There is no class overlap if $\gamma$ is 0. It is also worth noting that if $K$ is not a multiple of $T$, some classes may be exposed once less than the others. However, our experiments show that this does not significantly affect ensemble performance.

## C    WHY ADAPTING TEACHER LABEL PRIOR BEFORE AGGREGATION

### C.1    TEACHER PREDICTION ON LABEL PRIOR SHIFT

In supervised learning, a classifier parameterized by $\boldsymbol{\theta}$ tries to sample a correct label $y$ on the input $x$ by directly estimating conditional distribution $p(y|x; \boldsymbol{\theta})$. In our online multi-head learning, each teacher parameterized $\boldsymbol{\theta}_t$ learns corresponding true distribution $p_t(x, y) = p(x|y)p_t(y)$ whose label prior is differently class-skewed $p_t(y) \neq p_{t'}(y)$. In Bayes rule, the empirical inference over parameters $\boldsymbol{\theta}_t$ given specialty dataset $\mathcal{D}_t = \{(x_i, y_i)\}_{i=1}^{M}$ on each class-imbalanced distribution is as follows:

$$
p(\boldsymbol{\theta}_t | \mathcal{D}_t) \propto p(\boldsymbol{\theta}_t) \prod_{i=1}^{M} p(y_i | x_i; \boldsymbol{\theta}_t).
\tag{24}
$$

For an unknown data $(x_{M+1}, y_{M+1})$, the predictive distribution is marginalized over posterior distribution $p(\boldsymbol{\theta}_t | \mathcal{D}_t)$:

$$p(y_{M+1}|x_{M+1}; \mathcal{D}_t) = \int p(y_{M+1}|x_{M+1}; \boldsymbol{\theta}_t) p(\boldsymbol{\theta}_t | \mathcal{D}_t) d\boldsymbol{\theta}_t \tag{25}$$

Therefore, each teacher's prediction of given data is likely related to its class-skewed distribution.

### C.2 BIASED ACCURACY ON LABEL PRIOR SHIFT

In this section, we further discuss the class-imbalance causes the label prior shift can result in incorrect accuracy (Tian et al., 2020) when especially teacher's label prior has varying degrees of imbalance on uniform (student) label prior $p_t(y) \neq p(y)$. For the simplicity, we assume a teacher classifier as $f(x; \boldsymbol{\theta}_t) : \mathbb{R}^D \to \{0, 1\}^K$ on the $K$-way one-hot classification on $D$-dimensional inputs.

$$\text{Acc}(x, y) = \frac{1}{N} \sum_{k=1}^{K} \sum_{i=1}^{N} \mathbb{I}(f(x_i; \boldsymbol{\theta}_t) = k, y_i = k) = \sum_{k=1}^{K} \frac{N_k}{N} \left( \frac{1}{N_k} \sum_{i=1}^{N} \mathbb{I}(f(x_i; \boldsymbol{\theta}_t) = k, y_i = k) \right)$$

$$= \sum_{k=1}^{K} p(y = k) \text{Agreement}(y = k) = \mathbb{E}_{y \sim p(y)}[\text{Agreement}(y)]. \tag{26}$$

As shown in Eq. 26, the accuracy is equal to the expectation of agreement underlying the given label prior. When $p_t(y) \neq p(y)$, training with imbalanced data maximizes accuracy on $p_t(y)$ where majority classes are likely to observe. On the other hand, the accuracy of uniform data calculates the expectation of agreement on $p(y)$. Therefore, training in $p_t(y)$ is prone to bias towards large classes to maximize Eq. 26 and thus may result in inaccurate evaluation on uniform $p(y)$.

### C.3 LIKELIHOOD RELAXATION

As shown in Eq. 25 and Eq. 26, training imbalanced implies that a given teacher cannot be accurate in the minority data and allows teacher prediction to be closely related to its corresponding labels. Ren et al. (2020) introduce each negative-log likelihood (NLL) error of minority classes that should be adjusted more. They propose a manual relaxation method over the class-wise NLL by posing a discriminative "margin" denoted as $\gamma_k$ where $k$ is a class index. By carefully revisiting Theorem 2 in both Ren et al. (2020); Kakade et al. (2008), we discuss how we can quantitatively set the margin $\gamma_k$.

Suppose $\xi \geq 0$ is any threshold and $\mathcal{L}_t(\boldsymbol{\theta}_t)$ is the standard NLL in Softmax regression of our teachers on the class-imbalanced dataset. Denoting $\Omega_k$ is a subset of $k$-class, let $err_k(\xi)$ be zero-one loss from empirical samples in $k$-class subset: $err_k(\xi) = Pr_{(x,y) \in \Omega_k}[\mathcal{L}_t(\boldsymbol{\theta}_t) > \xi]$. In addition, we define $err_{\gamma,k}(\xi)$ is the zero-one $\gamma$-margin loss from empirical samples in $k$-class subset: $err_{\gamma,k}(\xi) = Pr_{(x,y) \in \Omega_k}[\mathcal{L}_t(\boldsymbol{\theta}_t) + \gamma_k > \xi]$.

**Theorem 2. (Ren et al., 2020)** *Assume that $\mathcal{L}_t$ is Lipschitz continuous and $\sup_{(x,y) \in \Omega} |\mathcal{L}_t(\boldsymbol{\theta}_t) - \xi| \leq C$ where $\Omega$ is an entire dataset. For any $\delta > 0$ with probability at least $1 - \delta$ over the samples, $\forall \gamma_k > 0$ and $\forall f \in \mathcal{F}$ in Theorem 2 of Kakade et al. (2008), neglecting empirical noise, we have*

$$err_k(\xi) \leq err_{\gamma,k}(\xi) + \frac{4\mathcal{R}_k(\mathcal{F})}{\gamma_k} + \sqrt{\frac{\log(\log_2 \frac{4C}{\gamma_k})}{n_k}} + \sqrt{\frac{\log(1/\delta)}{2n_k}} \tag{27}$$

where $\mathcal{R}_k(\mathcal{F})$ is the Rademacher complexity of a function family $\mathcal{F}$ (Kakade et al., 2008) and $n_k$ is the sample size of $k$-class subset. By discussion in Ren et al. (2020), we can have the relaxed generalization error bound $err_{uniform}(\xi)$ for the loss of uniformly class-distributed dataset.

$$err_{uniform}(\xi) \leq \frac{1}{K} \sum_{k=1}^{K} \left( err_{\gamma,k}(\xi) + \frac{4}{\gamma_k} \sqrt{\frac{\Gamma(\mathcal{F})}{n_k}} + \sqrt{\frac{\log(\log_2 \frac{4C}{\gamma_k})}{n_k}} + \sqrt{\frac{\log(1/\delta)}{2n_k}} \right) \tag{28}$$

where $\Gamma$ can be measured as a complexity of $\mathcal{F}$, following Thereom 3 of Kakade et al. (2008). To minimize the uniform error bound in Eq. 28 according to $n_k$, we should minimize the second term because the first term is a natural data loss and the other terms are negligible low-order losses.

With an equality constraint of $\sum_{k=1}^{K} \gamma_k = \rho$, we can solve the minimization problem of the second term by applying Cauchy-Schwarz inequality to get each optimal $k$-class margin $\gamma_k^*$.

$$\min \sum_{k=1}^{K} \frac{4}{\gamma_k} \sqrt{\frac{\Gamma(\mathcal{F})}{n_k}}, \quad \text{subject to} \quad \sum_{k=1}^{K} \gamma_k = \rho. \tag{29}$$

*Proof.* Given minimization problem can be written as

$$\min \sum_{k=1}^{K} \gamma_k \sum_{k=1}^{K} \frac{4}{\gamma_k} \sqrt{\frac{\Gamma(\mathcal{F})}{n_k}}. \tag{30}$$

By Cauchy-Schwarz inequality,

$$\sum_{k=1}^{K} \gamma_k \sum_{k=1}^{K} \frac{4}{\gamma_k} \sqrt{\frac{\Gamma(\mathcal{F})}{n_k}} \geq \left( \sum_{k=1}^{K} \gamma_k \cdot \frac{4}{\gamma_k} \sqrt{\frac{\Gamma(\mathcal{F})}{n_k}} \right)^2 = \left( \sum_{k=1}^{K} 4 \sqrt{\frac{\Gamma(\mathcal{F})}{n_k}} \right)^2. \tag{31}$$

Both sides are equal if and only if $\gamma_k$ and $\frac{4}{\gamma_k} \sqrt{\frac{\Gamma(\mathcal{F})}{n_k}}$ are linearly dependent. Thus, we choose a multiplier $\zeta^2$ for ease of calculation. Then, we have

$$\gamma_k = \zeta^2 \frac{4}{\gamma_k} \sqrt{\frac{\Gamma(\mathcal{F})}{n_k}}; \quad \gamma_k^2 = 4\zeta^2 \sqrt{\frac{\Gamma(\mathcal{F})}{n_k}}; \quad \gamma_k = 2\zeta \left( \frac{\Gamma(\mathcal{F})}{n_k} \right)^{1/4}. \tag{32}$$

Substitute $\gamma_k$ of Eq. 32 with those of the equality constraint in Eq. 29.

$$\rho = \sum_{k=1}^{K} \gamma_k = \sum_{k=1}^{K} 2\zeta \left( \frac{\Gamma(\mathcal{F})}{n_k} \right)^{1/4}; \quad \rho = 2\zeta \sum_{k=1}^{K} \left( \frac{\Gamma(\mathcal{F})}{n_k} \right)^{1/4}, \tag{33}$$

$$\zeta = \frac{\rho}{2 \sum_{k=1}^{K} \left( \frac{\Gamma(\mathcal{F})}{n_k} \right)^{1/4}}; \quad \frac{\gamma_k}{2 \left( \frac{\Gamma(\mathcal{F})}{n_k} \right)^{1/4}} = \frac{\rho}{2 \sum_{k=1}^{K} \left( \frac{\Gamma(\mathcal{F})}{n_k} \right)^{1/4}}, \tag{34}$$

$$\gamma_k = \frac{2\rho \left( \frac{\Gamma(\mathcal{F})}{n_k} \right)^{1/4}}{2 \sum_{k=1}^{K} \left( \frac{\Gamma(\mathcal{F})}{n_k} \right)^{1/4}} = \frac{2\rho \Gamma(\mathcal{F})^{1/4} \left( \frac{1}{n_k} \right)^{1/4}}{2\Gamma(\mathcal{F})^{1/4} \sum_{k=1}^{K} \left( \frac{1}{n_k} \right)^{1/4}}, \tag{35}$$

Finally, the optimal margin of the $k$-class subset, $\gamma_k^*$, is as follows:

$$\therefore \gamma_k^* = \frac{\rho n_k^{-1/4}}{\sum_{k=1}^{K} n_k^{-1/4}}. \tag{36}$$

$\square$

As a result of Eq. 36, $\gamma_k^*$ implies that independent margins are necessary according to $n_k$. Thus, minority classes sometimes require larger margins to be generalized. To make a uniform generalization error against each teacher prediction, denoted as Eq. 25, each teacher necessitates manually relaxing $k$-class NLL loss by adjusting Softmax outputs.

Corollary 2.1 of Ren et al. (2020) introduces that we can get the desired NLL loss from a sum of class-wise NLL loss and the given optimal margin. The straightforward derivation results in a compensating method for the $k$-class logit value. Let a conditional distribution by adapted logit values on each $t$-th teacher be $\hat{p}_t(y|x; \boldsymbol{\theta}_t)$. Then, we can define

$$\hat{p}_t(y = k|x; \boldsymbol{\theta}_t) = \frac{\exp(z_t[k] - \log \gamma_{t,k}^*)}{\sum_{k'=1}^{K} \exp(z_t[k'] - \log \gamma_{t,k'}^*)} = \frac{n_{t,k}^{\frac{1}{4}} \exp(z_t[k])}{\sum_{k'=1}^{K} n_{t,k'}^{\frac{1}{4}} \exp(z_t[k'])} \tag{37}$$

where $\gamma_{t,k}^*$ and $n_{t,k}$ denote each optimal $k$-class margin and size of $t$-th teacher and each $k$-class logit value $z_t[k]$ is compensated as much as $-\log \gamma_{t,k}^*$. However, Ren et al. (2020) suggests that since

Eq. 28 is not tight, a power of $1/4$ for $n_k$ becomes not powerful condition than using 1. Therefore, we can redefine

$$\hat{p}_t(y = k|x; \boldsymbol{\theta}_t) = \frac{n_{t,k} \exp(z_t[k])}{\sum_{k'=1}^{K} n_{t,k'} \exp(z_t[k'])} = \frac{\frac{n_{t,k}}{n} \exp(z_t[k])}{\sum_{k'=1}^{K} \frac{n_{t,k'}}{n} \exp(z_t[k'])}$$
$$= \frac{\tilde{p}_t(y = k) \exp(z_t[k])}{\sum_{k'=1}^{K} \tilde{p}_t(y = k') \exp(z_t[k'])}, \tag{38}$$

where $\tilde{p}_t(y = k)$ denotes class-imbalanced probability of $t$-th teacher and $n$ is the total number of dataset $\Omega$. Using $p(y = k)$, which is a uniform probability of $k$-class, then, we can have

$$\hat{p}_t(y = k|x; \boldsymbol{\theta}_t) = \frac{\tilde{p}_t(y = k) \exp(z_t[k])}{\sum_{k'=1}^{K} \tilde{p}_t(y = k') \exp(z_t[k'])} = \frac{\frac{\tilde{p}_t(y=k)}{p(y=k)} \exp(z_t[k])}{\sum_{k'=1}^{K} \frac{\tilde{p}_t(y=k')}{p(y=k')} \exp(z_t[k'])}$$
$$= \frac{\exp(z_t[k] + \log(\frac{\tilde{p}_t(y=k)}{p(y=k)}))}{\sum_{k'=1}^{K} \exp(z_t[k'] + \log(\frac{\tilde{p}_t(y=k')}{p(y=k')}))} = \frac{\exp(z_t[k] - \log(\frac{p(y=k)}{\tilde{p}_t(y=k)}))}{\sum_{k'=1}^{K} \exp(z_t[k'] - \log(\frac{p(y=k')}{\tilde{p}_t(y=k')}))}$$
$$= \frac{\exp(z_t[k] - \log(\frac{1}{\kappa_t(y=k)}))}{\sum_{k'=1}^{K} \exp(z_t[k'] - \log(\frac{1}{\kappa_t(y=k')}))} \tag{39}$$

where $\kappa_t(y = k)$ is our proposed CR function over $t$-th teacher. Eq. 39 is, as a result, the same as Eq. 7, PC-Softmax (Hong et al., 2021). Thus, we little adjust each $k$-class logit value $z_t[k]$ as much as $-\log(1/\epsilon)$ in this paper.

## D  DIVERSITY: AVERAGED PAIRWISE JENSEN-SHANNON DIVERGENCE

We measure the diversity of teacher outputs based on Jensen-Shannon Divergence (JSD), which assesses how similar two distributions are. The two distributions are mutually informative if the diversity is zero. Given the $i$-th sample, we formulate the diversity among $T$ teachers as follows:

$$Diversity = \frac{1}{N} \sum_{i=1}^{N} Div_i; \tag{40}$$

$$Div_i = \frac{1}{T(T-1)} \sum_{t \in [1,T]} \sum_{t' \in [1,T] \setminus t} \frac{1}{2} \left[ D_{KL}(p_t^i \| \sigma) + D_{KL}(p_{t'}^i \| \sigma) \right], \tag{41}$$

where $p_t^i$ is the output probability distribution of $t$-th teacher and $\sigma = (p_t^i + p_{t'}^i)/2$. For our proposed method, probability distributions after post-compensation are used.

## E  EXPERIMENTAL SETTINGS

### E.1  EXPERIMENTAL CONFIGURATIONS

**Datasets.**   We compare our proposed method to previous online KD works using three datasets. CIFAR-10 and CIFAR-100 datasets (Krizhevsky, 2009) each have 50K training images and 10K test images, with each image belonging to one of 10 or 100 classes. ImageNet (Deng et al., 2009) contains 1.2M training and 50K validation images in 1K classes.

**Training settings.**   For CIFAR datasets, we train all models for 300 epochs. We use SGD with a momentum of 0.9. The learning rate begins at 0.1 and decreases by one-tenth every 150 and 225 epochs. We employ a standard data augmentation strategy from He et al. (2016) and normalize all images by each channel mean and standard deviation. The batch size is set to 128, and the weight decay to $5 \times 10^{-4}$. The ramp-up period $\alpha$ of a balancing factor $\lambda(e)$ for student knowledge distillation loss is 80, where $e$ is an epoch. During the first 80 epochs, $\lambda(e)$ varies from 0 to 1. We perform a grid search to find each model's optimal exposure $\epsilon$. We find a best $\epsilon$ among $[0.1, 0.3, 0.5, 0.8, 1.0]$

by measuring ERR, and choose 0.5 for ResNet-32, 0.3 for ResNet-110, 0.5 for DenseNet-40-12, 0.3 for EfficientNetB0, and 0.8 for MobileNetV2. We choose different exposure for models as they differ in deep network architecture and the ratio of peer/shared parameters, as shown in Section E.2. All parameters are initialized with MSRA initialization (He et al., 2015). To compare our method with previous works, we use the officially released implementation code[234] for the works and three evaluation metrics on each method is fairly measured with the training settings above. While we use $\tau = 3$ for knowledge distillation temperature, DML uses $\tau = 1$, and CLILR uses $\tau = 2$; those values are reported in the original paper.

We train all ImageNet models for 90 epochs. The learning rate begins at 0.1 and decreases by one-tenth at the 30 and 60 epochs. The mini-batch size is set to 256, and the weight decay to $1 \times 10^{-4}$. A balancing factor $\lambda(e)$ has a ramp-up period $\alpha$ of 20 where $e$ is an epoch. For our knowledge distillation, we set $\tau = 3$. For all models, the exposure $\epsilon$ of 0.7 is used. We also used MSRA initialization for ImageNet.

### E.2 ARCHITECTURAL CONFIGURATIONS OF THE PEER-BASED METHOD

We separate the shared and teacher-specific parts from the start of the last block to build a peer-based architecture for various deep models on both CIFAR datasets. For this purpose, we adhere to the strategy in Chen et al. (2020). We divide the shared part from the teacher-specific part at the beginning of the last two building blocks for all methods on MobileNetV2 and EfficientNetB0 built for more comparisons in this paper. As a result, network-based models have far more parameters than peer-based models, as shown in Table 3. We also follow the separating strategy in Chen et al. (2020) for ResNet on the ImageNet dataset; we split the last two residual blocks to build peer-based architecture.

Table 3: The pure parameter ratio of DNNs. *Network-based / Peer-based* denotes the parameter ratio of network-based models to peer-based models. *Peer / Shared* denotes the parameter ratio of single peer head to the shared part. The models are in order of Table 1. This ratio excludes additional parameters by extra modules generated in our benchmark works.

| Dataset | Params. Ratio | ResNet-32 | ResNet-110 | DenseNet-40-12 | EfficientNetB0 | MobileNetV2 |
|---|---|---|---|---|---|---|
| CIFAR-10 | *Network-based / Peer-based* | 1.22 | 1.22 | 3.94 | 1.22 | 1.22 |
| | *Peer / Shared* | 3.15 | 3.17 | 0.01 | 3.22 | 3.12 |
| CIFAR-100 | *Network-based / Peer-based* | 1.22 | 1.21 | 3.54 | 1.21 | 1.21 |
| | *Peer / Shared* | 3.20 | 3.25 | 0.05 | 3.25 | 3.33 |

**Diversity disparity among various model architectures.** Accepting that DNNs are architecturally distinct, we can empirically analyze through Figure 3, and Table 1 that deriving diversity can be significantly difficult if the amounts of peer-head parameters are considerably smaller than the shared part. DenseNet-40-12, for example, has almost all shared parameters because this architecture uses the teacher-specific part as only a fully-connected layer (Chen et al., 2020). Thus, we can speculate that only minor individual parameters are included for specialization, implying that diversity is possible (outperforms the previous methods), but specialization remains difficult. Therefore, The diversity of our proposed method on DenseNet-40-12 is lower than that of other models. Furthermore, we investigate why MobileNetV2 and EfficientNetB0 have less diversity than ResNet-32. Despite having a similar *Peer/Shared* parameter ratio, the aforementioned structural differences can be caused by an intermediate layer type, e.g., spatial or depthwise-separable convolution; however, concrete analysis is our future topic.

## F SUPPLEMENTARY RESULTS

### F.1 COMPARISON WITH NETWORK-BASED METHODS

To make a fair comparison with DML and a network-based variant of OKDDip, we rebuilt our framework as a network-based one. From a results of ECE in Table 4 and Table 1, network-

---

[2]https://github.com/DefangChen/OKDDip-AAAI2020
[3]https://github.com/Lan1991Xu/ONE_NeurIPS2018
[4]https://github.com/Jangho-Kim/FFL-pytorch

based online KD shows more effectiveness than peer-based in producing a more calibrated student. Regardless of class size, our method outperforms previous works in ResNet-110, EfficientNetB0, and MobileNetV2. On CIFAR-10, our method outperforms in ResNet-32 and DenseNet-40-12, but falls short of OKDDip in ERR on CIFAR-100. However, ours is still better than OKDDip, about 2x in ECE. As a result, our student is more accurately confident than OKDDip.

Table 4: The generalization comparison with previous network-based methods on the student model. ERR and ECE use a percentage (%), and NLL is a loss value. Thus, the lower it is, the better. The numbers are the test results of three random experiments and filled in the mean(±std). The best result within each type is indicated in bold.

| Dataset | Method | ResNet-32 ERR | ResNet-32 ECE | ResNet-32 NLL | ResNet-110 ERR | ResNet-110 ECE | ResNet-110 NLL | DenseNet-40-12 ERR | DenseNet-40-12 ECE | DenseNet-40-12 NLL | EfficientNetB0 ERR | EfficientNetB0 ECE | EfficientNetB0 NLL | MobileNetV2 ERR | MobileNetV2 ECE | MobileNetV2 NLL |
|---|---|---|---|---|---|---|---|---|---|---|---|---|---|---|---|---|
| CIFAR-10 | DML | 6.01 (±0.15) | 3.06 (±0.10) | 0.22 (±0.00) | 5.63 (±0.26) | 2.14 (±0.05) | 0.19 (±0.00) | 6.50 (±0.10) | 2.28 (±0.18) | 0.21 (±0.00) | 8.05 (±0.18) | 1.80 (±0.46) | 0.25 (±0.01) | 10.35 (±0.20) | 1.21 (±0.06) | 0.30 (±0.00) |
| | OKDDip | 5.72 (±0.02) | 3.71 (±0.14) | 0.24 (±0.00) | 4.45 (±0.14) | 2.47 (±0.03) | 0.17 (±0.00) | 5.94 (±0.05) | 2.80 (±0.16) | 0.21 (±0.00) | 7.64 (±0.07) | 2.98 (±0.15) | 0.25 (±0.00) | 9.87 (±0.07) | 1.93 (±0.28) | 0.30 (±0.00) |
| | KDCL | 6.04 (±0.16) | 3.21 (±0.70) | 0.22 (±0.01) | 5.07 (±0.12) | 2.20 (±0.64) | 0.18 (±0.01) | 6.12 (±0.18) | 1.97 (±0.20) | 0.19 (±0.00) | 7.95 (±0.11) | 1.59 (±0.37) | 0.24 (±0.00) | 10.75 (±0.07) | 1.51 (±0.16) | 0.33 (±0.00) |
| | Ours | **5.68** (±0.16) | **2.32** (±0.22) | **0.19** (±0.00) | **4.40** (±0.20) | **2.06** (±0.17) | **0.16** (±0.00) | **5.89** (±0.15) | **1.90** (±0.21) | **0.19** (±0.00) | **7.64** (±0.08) | **1.56** (±0.22) | **0.23** (±0.00) | **9.80** (±0.24) | **0.99** (±0.21) | **0.29** (±0.00) |
| CIFAR-100 | DML | 26.22 (±0.15) | 4.77 (±0.13) | 0.93 (±0.00) | 22.83 (±0.53) | 7.85 (±0.25) | 0.88 (±0.00) | 27.05 (±0.18) | 2.82 (±0.38) | 0.93 (±0.00) | 27.78 (±0.10) | 3.40 (±0.33) | 0.97 (±0.00) | 31.76 (±0.08) | 2.73 (±0.92) | 1.10 (±0.00) |
| | OKDDip | **25.46** (±0.04) | 7.43 (±0.65) | 0.96 (±0.01) | 21.44 (±0.33) | 9.22 (±0.44) | 0.86 (±0.01) | **26.25** (±0.38) | 3.88 (±0.70) | 0.91 (±0.00) | 26.68 (±0.15) | 7.93 (±0.70) | 1.00 (±0.04) | 31.56 (±0.22) | 2.94 (±0.30) | 1.10 (±0.00) |
| | KDCL | 25.55 (±0.37) | **1.85** (±0.62) | 0.90 (±0.01) | 22.75 (±1.14) | **3.11** (±0.58) | **0.80** (±0.02) | 26.67 (±0.12) | 1.66 (±0.10) | 0.91 (±0.00) | 26.81 (±0.19) | 3.49 (±1.34) | 0.95 (±0.02) | 31.37 (±0.26) | 1.45 (±0.27) | 1.09 (±0.00) |
| | Ours | 25.52 (±0.10) | 3.81 (±0.39) | **0.90** (±0.00) | **21.44** (±0.11) | 7.22 (±0.25) | 0.82 (±0.01) | 26.29 (±0.11) | **1.66** (±0.15) | **0.91** (±0.00) | **26.66** (±0.24) | **2.59** (±0.21) | **0.94** (±0.00) | **31.14** (±0.31) | **1.43** (±0.12) | **1.09** (±0.00) |

## F.2 DIVERSITY CHANGE ON THE VARIATION OF RAMP-UP PERIOD

This section shows how our diversity is large and maintained well throughout training according to the variation of ramp-up period $\alpha$. In the online KD works, $\alpha$ has been used to modulate the power of KD strength to control homogenization. For example, when $\alpha$ is 80, $\lambda(e)$ in Eq 13 varies from 0 to 1 during the first 80 epochs. As shown in Figure 6, CLILR, FFL-S, and ONE are sensitive to the variation of $\alpha$. In particular, when $\alpha$ is small, the previous works have suffered from homogenization since early epochs. However, OKDDip and ours have not been affected by the variation of $\alpha$. In addition, our method consistently exhibits the highest diversity over previous works.

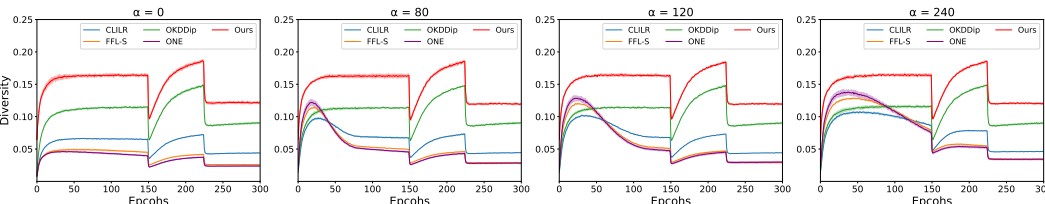

Figure 6: Diversity comparisons during entire training time for ResNet-32 on CIFAR-100. The shaded region represents the mean(±std), calculated from three trials. We plot each diversity based on PC-Softmax (**ours**) and Softmax (**others**) while using training set.

## F.3 VISUALIZATION OF TEACHER POSTERIOR DISTRIBUTION

This section depicts how the teacher's output varies when predicting specific class samples for each $t$-th teacher. When $T = 4$, we apply two deep neural networks, ResNet-32 and ResNet-110, on CIFAR-100 and test on exposure $\epsilon \in [0.1, 0.3, 1.0]$. We create an arbitrarily partial test set $\tilde{\mathcal{D}}$ including only the specific labeled data as $\tilde{\mathcal{Y}} = [1, 24]$; we directly generate the skewed label distribution with the number of samples equal within $\tilde{\mathcal{Y}}$ and zero in the reset of classes.

For each sample $(x_i, y_i) \sim \tilde{\mathcal{D}}$, we can get averaged predictions for each teacher. We first define the conditional distribution $p_t(y|x_i; \boldsymbol{\theta}_t)$ in $K$-class Softmax, which can be represented as a multinomial distribution:

$$p_t(y|x_i; \boldsymbol{\theta}_t) = \prod_{k=1}^{K} p_t(y = k|x_i; \boldsymbol{\theta}_t)^{\mathbf{1}\{y=k\}}; \; p_t(y = k|x_i; \boldsymbol{\theta}_t) = \frac{\exp(z_t^i[k])}{\sum_{k'=1}^{K} \exp(z_t^i[k'])}, \; t \leq T, \quad (42)$$

where $\mathbf{1}\{\cdot\}$ denotes the indicator function. The logit of class $k$ given an $i$-th input sample $(x_i, y_i)$ produced by the $t$-th teacher model is denoted by $z_t^i[k]$. Second, we take the conditional distributions for each $t$-th teacher and average them across all samples on $\tilde{\mathcal{D}}$.

$$p_t(y|x; \boldsymbol{\theta}_t) = \frac{1}{N} \sum_{i=1}^{N} p_t(y|x_i; \boldsymbol{\theta}_t), \quad t \leq T, \tag{43}$$

where $N$ denotes the total number of samples in $\tilde{\mathcal{D}}$. To demonstrate the post-compensation (PC) effect, we adapt the original label prior as introduced in Section 3.5 and thus replace $p_t(y|x_i; \boldsymbol{\theta}_t)$ with $\hat{p}_t(y|x_i; \boldsymbol{\theta}_t)$.

In Figure 7, we plot Eq 43 and a variant of Eq 43 replaced by $\hat{p}_t(y|x; \boldsymbol{\theta}_t)$. Smaller $\epsilon$ leads to different conditional distributions when a model is slimmer, inducing specialization and dramatic diversity. After applying the PC strategy, we can see that teachers still maintain diversity in the uniform distribution.

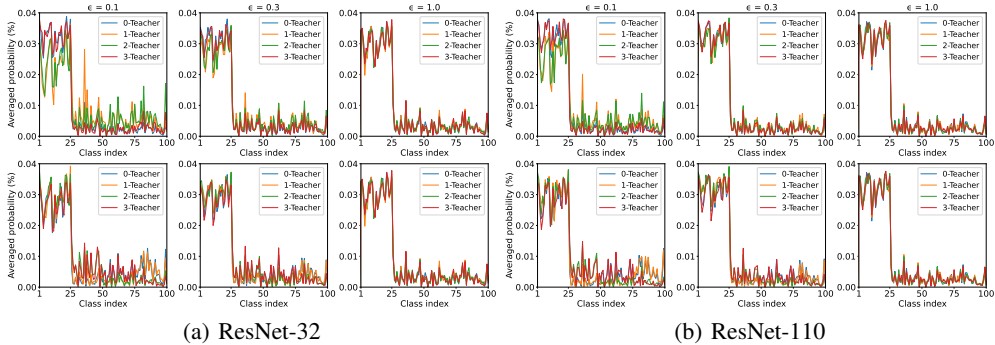

(a) ResNet-32          (b) ResNet-110

Figure 7: Visualization of averaged teachers' posterior distribution on the specific labeled dataset $\hat{\mathcal{D}}$. The front number of a teacher is a teacher index. We plot each averaged conditional distribution based on Softmax (**up**) and PC-Softmax (**down**).

### F.4    Ensemble Confidence Visualization

As shown in Figure 8, we visualize ensemble confidence compared to previous methods on both positive and negative samples. As shown in Figure 8, our ensemble is less over-confident for positive samples than previous methods. Especially, our ensemble more confidently mispredicts for negative samples. It implies that our ensemble may have a lower chance of miscalibrated failure (being completely incorrect) than the others. That is, our ensemble experiences failure uncertainly.

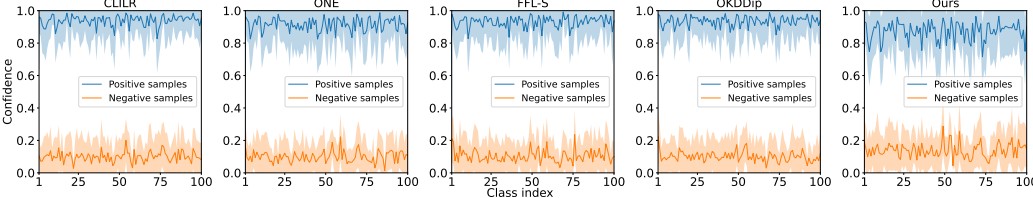

Figure 8: Confidence of an ensemble posterior distribution corresponds to each class. For each $k$ class, *positive samples* denote the correct samples corresponding class $k$, and *negative samples* denote the incorrect samples on the class $k$. The shaded area corresponds to the mean($\pm$std).

## G    Limitation

Our ensemble method produces a better confidence calibration by leveraging two key factors: combining probabilities and teacher diversity. To resolve label prior shift among teachers and match the

student label distribution, the class probabilities are individually post-compensated. PC is necessary, but naively employing the PC strategy can result in sometimes overbalanced posterior probabilities on the rest of the specialty classes (Ren et al., 2020). We studied that generalization error bound for minority classes with fewer samples should be carefully considered in Appendiex C.3; we theoretically discussed that tightness to derive the post-compensation ratio is sometimes well-assumed from Eq. 28. We empirically discovered that our framework suffers from the overbalanced problem when re-scaling teacher outputs on the rest of the specialty labels before forming an ensemble. We conjecture that an estimator derived from importance sampling can have inherent difference from an estimator derived from basic Monte-Carlo sampling of the actual imbalanced joint distribution; thus, we speculate that a degree of experience with the out of the specialty classes will be a little different from the actual situation. In future work, we will investigate the posterior overbalanced problem after the PC strategy more thoroughly and fundamentally to obtain better predictions.

## H  POTENTIAL SOCIETAL IMPACT

This work has the same potential impact as any neural network compression study. The positive effect first comes from reducing the resource overhead of deep learning models during inference time. Second, a compressed model with only essential knowledge has more potential because it achieves comparable performance with less power and can even exhibit better generalization than the larger capacity model. Therefore, we can deploy neural network models to mobile phones or edge devices, expecting acceptable performance. We thus take a step closer to energy-friendly deep learning, facilitating a wider use of Artificial Intelligence in industrial IoT or smart home technology.

At the same time, research on neural network compression may have some negative consequences. For example, if neural network models are more widely used for wearable devices or surveillance cameras, privacy invasion or cybercrime is possible. In addition, the malfunction of industrial IoT devices could cause a severe problem for the whole production process.

