# OpenReview forum: "Enriching Online Knowledge Distillation with Specialist Ensemble"
_ICLR.cc/2023/Conference — Submitted to ICLR 2023_

### Official Review · Reviewer_HD42 · 2022-10-24

**Confidence:** 4
**Clarity, Quality, Novelty And Reproducibility:** Please refer to my detailed comments …
**Correctness:** 1
**Technical Novelty And Significance:** 2
**Empirical Novelty And Significance:** 2
**Recommendation:** 3

**Strength And Weaknesses:**

Strength:

(1) The motivation is clear, where this paper mainly aims to solve the issue of diverse teachers' construction.

(2) The organization is good, which makes the paper easy to follow.

(3) The authors also provide detailed proof for theorems and corollaries.

(4) Sufficient experimental analysis is given for better understanding.

Weaknesses:

(1) My main concern lies in the experimental results. Several methods achieve much better results than the proposed method on all these datasets. The authors compared with PCL on ImageNet, and it achieves a 0.05% improvement over PCL. I wonder why the authors do not compare with it on CIFAR10/100. According to the results on PCL, the results on CIFAR under various backbones are better than the proposed method. Moreover, one important reference[1] is missing, which achieves much better results than the proposed method, including ImageNet.

(2) As introduced by the authors, label prior shift has been extensively discussed by existing methods. Please analyze the difference between the adopted strategy and existing methods. Is this a simple combination with KD?

(3) Sampling has also been well studied. Please also compare the importance sampling with existing methods.

(4) Based on the above analysis, I think the novelty could be improved.

[1] MULTI-VIEW CONTRASTIVE LEARNING FOR ONLINE KNOWLEDGE DISTILLATION, ICASSP, 2021


**Summary Of The Paper:**

Towards the ensemble-based KD, this paper first presents a label prior shift to induce evident diversity among the same teachers. Then the authors propose an aggregation strategy that uses post-compensation in specialist outputs and conventional model averaging. Experiments on several datasets validate its effectiveness.

**Summary Of The Review:**

The presentation and writing are good. But the novelty could be improved. And the experimental results are not convincing.

---

> ### Author Response · Authors · 2022-11-18
> **Response to Reviewer HD42 (2/2)**
>
> > As introduced by the authors, label prior shift has been extensively discussed by existing methods. Please analyze the difference between the adopted strategy and existing methods. Is this a simple combination with KD?
>
> > Sampling has also been well studied. Please also compare the importance sampling with existing methods.
>
> One important aspect of online KD is that the peers share a mini-batch. Therefore, the training distributions of peers are fated to be the same. Our key idea is to break this limitation and expose peers to different training distributions so as to increase the diversity of peers. To achieve this seemingly infeasible goal, we first employ importance sampling to simulate different label prior shifts out of the same mini-batch samples. And then, we deal with this artificial label imbalance by adopting PC-Softmax. This compensation is necessary because the distribution each peer learns is elaborately made different from the original distribution.
> The bottom line is we do not argue that we made an advancement in label prior shift or importance sampling. Instead, we claim that using label prior shift and importance sampling in this context is a novel contribution to online KD research.
>
> We greatly appreciate the time and effort you spent reviewing our paper, which we are certain will make our work better.
>
> - Reference
>
> [1a] Guile Wu et al., Peer collaborative learning for online knowledge distillation. In Proceedings of the AAAI Conference on Artificial Intelligence, 2021.
>
> [1b] https://github.com/shaoeric/Peer-Collaborative-Learning-for-Online-Knowledge-Distillation.
>
> [2a] Chuanguang Yang et al., Multi-view contrastive learning for online knowledge distillation, IEEE International Conference on Acoustics, Speech, and Signal Processing (ICASSP), 2021.
>
> [2b] https://github.com/winycg/MCL-OKD/tree/5de042a5fbd4829f82caa68facc7586f59ea3732.

---

> ### Author Response · Authors · 2022-11-18
> **Response to Reviewer HD42 (1/2)**
>
> We would like to thank you for your valuable and constructive comments.
>
> Based on your concerns and questions, we have conducted the required experiments to further show the effectiveness of our proposed method.
>
> > Regarding experimental results
>
> Concerning the comparison with PCL [1a], as far as we know, PCL has yet to release an officially implemented code until we organize our experimental results. As a result, we could not fairly compare PCL on CIFAR-10/100 for the three measurements: ERR, ECE, and NLL. To thoroughly compare PCL, we use reproduced codes by a contributor [1b] for this response.
>
> Before proceeding with additional experiments, we will explain why our method shows a marginal improvement over PCL in the accuracy you are concerned about: PCL employs temporal mean networks (TMNs) and multiple peers and makes use of TMNs as the same as the number of peers. For instance, if the number of peers is two, PCL behaves as if the number of peers is four because TMNs include trainable parameters as much as the peers. As a result, PCL utilizes nearly double the resources to achieve comparable accuracy in training time to ours. Therefore, comparing PCL's accuracy with our T=2 is unfair. The accuracy of PCL should be compared by at least T=3 or T=4 of our method, which seems fair.
>
> Therefore, we conduct additional experiments when T=3 of our method to compare it fairly with PCL on ImageNet. In addition, we perform requested comparisons with PCL on CIFAR-10/100 to describe the efficacy of ERR (%), ECE (%), and NLL. Meanwhile, we do the additional evaluation with MVCL [2a], which you recommended as milestone work. We use their official code [2b] and the same training strategy described in Appendix.E. We match the number of peers, four for PCL and MVCL, for a fair comparison.
>
> The following are the CIFAR-10/100 results. The numbers are the test results of averaged three random experiments. We abbreviate each model as R32: ResNet-32, R110: ResNet-110, D40: DenseNet-40-12, EB0: EfficientNetB0, and MNV2: MobileNetV2.
>
> | Dataset |  Method | R32 ERR | R32 ECE | R32 NLL | R110 ERR | R110 ECE | R110 NLL | D40 ERR | D40 ECE | D40 NLL | EB0 ERR | EB0 ECE | EB0 NLL | MNV2 ERR | MNV2 ECE | MNV2 NLL |
> |:---------:|--------|:-----:|:-----:|:-----:|:-----:|:-----:|:-----:|:-----:|:-----:|:-----:|:-----:|:-----:|:-----:|:-----:|:-----:|:-----:|
> |  CIFAR | PCL    | 6.12        |  3.76   | 0.25 | 4.77 | 3.42 | 0.23 |  6.84 | 3.61| 0.25 | 7.12 | 3.81 | 0.25 | 11.35 | 3.08  |  0.35   |
> |     -10    | MVCL| 5.80        |   3.69  | 0.25 | 4.61 | 2.78 | 0.19 |  6.81 | 3.00 | 0.24 |  7.31 |3.44 |  0.26 |  11.44 | 2.99  |  0.35   |
> |              | Ours    | **5.61**   | **3.14** | **0.23** | **4.49**|  **2.29** | **0.17** | **6.78** | **2.82** | **0.24** | **7.08** | **2.55** | **0.24** | **11.27** | **2.00** | **0.34** |
> | CIFAR   | PCL     | 26.78      | 10.56    | 1.09 |  21.02 | 10.81 |  0.92 | 29.10 | 10.85 | 1.15 |  27.59  | 12.08  | 1.16  | 34.94  |  11.46  |  1.37 |
> |    -100   | MVCL | 26.16      |   9.40    | 1.01 |  21.30 | 9.97   |  0.90 | 28.85 |  6.04  | 1.04 |  27.83  |  10.77 |  1.15 |  32.93   |  4.34 |  1.16   |
> |              | Ours     | **25.68** | **5.30** | **0.93** | **20.94** | **6.68** |  **0.80** | **28.33** | **5.21** | **1.02** | **27.56** | **9.75** | **1.15** |  **32.41** | **3.01** | **1.13** |
>
> - On ImageNet, our top-1 ERR is 29.44% when T=3 on ResNet-18. Our method outperforms PCL by 0.14%. Furthermore, compared to MVCL for ResNet-34 on ImageNet, our method and MVCL improve upon each baseline by 1.31% and 0.79% (reported in [2a]), respectively, for top-1 ERR.
>
> In conclusion, on CIFAR-10/100 and ImageNet datasets, our method outperforms PCL and MVCL.
>
> The most recent manuscript includes PCL experimental results. If our paper is accepted, we will include all results, summarizing a discussion of MVCL in the camera-ready version.

---

> > ### Comment · Reviewer_HD42 · 2022-12-07
> > **Follow up**
> >
> > Thanks for the detailed response, especially the additional experimental results. However, the reported results of MVCL is not consistent with that reported in their original paper and released code. I guess that the reproduction within this short time may not achieve the optimal results. Besides, the simple ensemble operation in MVCL can also lead to much better results. And this paper is published one and a half years ago, which is also very simple. Moreover, I also agree with Reviewer Xcq2 that the novelty is not good enough. Based on the above observation, I still think the current paper is not ready to be published in this top conference.

---

> > > ### Author Response · Authors · 2022-12-13
> > > **Response to Reviewer HD42**
> > >
> > > Dear Reviewer, HD42,
> > >
> > >
> > > Thank you so much for your positive and constructive feedback. We appreciate the time and effort you have put into reviewing our modifications and responses.
> > >
> > > We would like to highlight the novelty of our method; it uses a label prior shift, the first work to the best of our knowledge, and an ensemble technique with post-compensated Softmax to enrich ensemble quality. The use of importance sampling makes our novel approach more reproducible and viable.
> > >
> > > The ensemble operation of MVCL is straightforward in that it averages the logits of each peer, which was traditionally accepted. As shown in Figure 4, we agree that this operation can result in highly accurate ensemble prediction. One of our novel contributions in this paper is to emphasize that model averaging with probability distributions, rather than logits, results in superior accuracy but lower calibration error.
> > > We would like to inform you that this paper recommends using it in conjunction with logarithms for knowledge distillation to a student model.
> > >
> > > Compared to MVCL, published long periods ago, our method is also easy to follow, as you have seen. Moreover, our method makes good accuracy and better-calibrated error, as shown in the experimental results of a reply (1/2); it is more predictable than MVCL.
> > >
> > >
> > > Sincerely,
> > >
> > > The authors

---

### Official Review · Reviewer_Xcq2 · 2022-10-25

**Confidence:** 3
**Clarity, Quality, Novelty And Reproducibility:** The paper is clear in most parts---so…
**Correctness:** 3
**Technical Novelty And Significance:** 2
**Empirical Novelty And Significance:** Not applicable
**Recommendation:** 6

**Strength And Weaknesses:**

The paper is fairly well-written and the high-level problem is interesting. The writing is clear and easy to follow. And the experiment part especially ablation study is well designed to show the improvement over previous methods.
However, there seems lack of comparison on CIFAR with more current works like CGL [1] or PCL [2]. Also there is a most recent one L-MCL [3] that has better performance. And the performance over those method seems not very significant. (It seems more common to report acc than error rate. Using error rate makes it hard to efficiently compare with the unlisted models.)

[1] Qiushan Guo, Xinjiang Wang, Yichao Wu, Zhipeng Yu, Ding Liang, Xiaolin Hu, and Ping Luo. Online
knowledge distillation via collaborative learning. In Proceedings of the IEEE/CVF Conference on
Computer Vision and Pattern Recognition (CVPR), 2020
[2] Guile Wu and Shaogang Gong. Peer collaborative learning for online knowledge distillation. In
Proceedings of the AAAI Conference on Artificial Intelligence, 2021.
[3] Yang C, An Z, Zhou H, et al. Online Knowledge Distillation via Mutual Contrastive Learning for Visual Recognition[J]. arXiv preprint arXiv:2207.11518, 2022.

**Summary Of The Paper:**

This paper proposes a online knowledge distillation framework with specialized ensembles. It first uses a label prior shift to generate different teachers for student to learn from. Then it uses PC-Softmax to post-compensate teacher logits and an averaged classifier manner to aggregate teacher predictions. And the experiments part show the improvement over previous method on CIFAR and ImageNet.

**Summary Of The Review:**

Overall, the paper proposes a new approach that uses specialized ensembles to do online KD. I would be happy to raise my score if the authors conducted the additional evaluations discussed in my review above.

---

> ### Author Response · Authors · 2022-11-18
> **Response to Reviewer Xcq2 (2/2)**
>
> - Reference
>
> [1a] Qiushan Guo, Xinjiang Wang, Yichao Wu, Zhipeng Yu, Ding Liang, Xiaolin Hu, and Ping Luo. Online knowledge distillation via collaborative learning. In Proceedings of the IEEE/CVF Conference on Computer Vision and Pattern Recognition (CVPR), 2020.
>
> [2a] Guile Wu and Shaogang Gong. Peer collaborative learning for online knowledge distillation. In Proceedings of the AAAI Conference on Artificial Intelligence, 2021.
>
> [3a] Chuanguang Yang, Zhulin An, Linhang Cai, and Yongjun Xu, Online Knowledge Distillation via Mutual Contrastive Learning for Visual Recognition, arXiv preprint arXiv:2207.11518, 2022.
>
> [1b] https://github.com/shaoeric/Online-Knowledge-Distillation-via-Collaborative-Learning
>
> [2b] https://github.com/shaoeric/Peer-Collaborative-Learning-for-Online-Knowledge-Distillation
>
> [3b] https://github.com/winycg/MCL

---

> ### Author Response · Authors · 2022-11-18
> **Response to Reviewer Xcq2 (1/2)**
>
> We would like to thank you for your valuable and constructive comments.
>
> Based on your suggestions, we have conducted additional experiments on the three milestone works (CGL [1a], PCL [2a], and MCL [3a]). Before we discuss the results, we would like to notify you that no official implementation code for the CGL (denoted as KDCL in our manuscript) and PCL has been released. As a result, we could not compare CGL and PCL fairly on CIFAR-10/100 when we wrote the manuscript. Thus, we use reproduced codes for CGL [1b] and PCL [2b], and we use an official code for L-MCL [3b] to do the requested experiments.
>
> We classified three works for a fair comparison; PCL and L-MCL are compared with peer-based methods, and CGL is compared with network-based methods, which was provided as our additional comparison in Appendix F.1. We used the same training strategy described in Appendix E and matched the number of peers, four for all the methods.
>
> We fairly evaluated all the methods on the three measurements (ERR (%), ECE (%), and NLL). The numbers are the test results of averaged three random experiments. We abbreviate each model as R32: ResNet-32, R110: ResNet-110, D40: DenseNet-40-12, EB0: EfficientNetB0, and MNV2: MobileNetV2. The results are as follows:
>
> | Dataset |  Method | R32 ERR | R32 ECE | R32 NLL | R110 ERR | R110 ECE | R110 NLL | D40 ERR | D40 ECE | D40 NLL | EB0 ERR | EB0 ECE | EB0 NLL | MNV2 ERR | MNV2 ECE | MNV2 NLL |
> |:---------:|--------|:-----:|:-----:|:-----:|:-----:|:-----:|:-----:|:-----:|:-----:|:-----:|:-----:|:-----:|:-----:|:-----:|:-----:|:-----:|
> |  CIFAR | PCL    | 6.12        |  3.76   | 0.25 | 4.77 | 3.42 | 0.23 |  6.84 | 3.61| 0.25 | 7.12 | 3.81 | 0.25 | 11.35 | 3.08  |  0.35   |
> |     -10    | L-MCL| 6.78        |   3.82  | 0.27 | 5.42 | 2.87 | 0.20 |  7.25 | 2.89 | 0.24 |  8.42 |3.08 |  0.29 |  13.01 | 2.34  |  0.39   |
> |              | Ours    | **5.61**   | **3.14** | **0.23** | **4.49**|  **2.29** | **0.17** | **6.78** | **2.82** | **0.24** | **7.08** | **2.55** | **0.24** | **11.27** | **2.00** | **0.34** |
> | CIFAR   | PCL     | 26.78      | 10.56    | 1.09 |  21.02 | 10.81 |  0.92 | 29.10 | 10.85 | 1.15 |  27.59  | 12.08  | 1.16  | 34.94  |  11.46  |  1.37 |
> |    -100   | L-MCL | 27.23      |   7.30    | 1.00 |  21.12 | 7.43   |  0.81 | 29.81 |  5.50  | 1.07 |  28.36  |  10.39 |  1.15 |  34.83   |  5.45 |  1.24   |
> |              | Ours     | **25.68** | **5.30** | **0.93** | **20.94** | **6.68** |  **0.80** | **28.33** | **5.21** | **1.02** | **27.56** | **9.75** | **1.15** |  **32.41** | **3.01** | **1.13** |
>
> | Dataset |  Method | R32 ERR | R32 ECE | R32 NLL | R110 ERR | R110 ECE | R110 NLL | D40 ERR | D40 ECE | D40 NLL | EB0 ERR | EB0 ECE | EB0 NLL | MNV2 ERR | MNV2 ECE | MNV2 NLL |
> |:---------:|--------|:-----:|:-----:|:-----:|:-----:|:-----:|:-----:|:-----:|:-----:|:-----:|:-----:|:-----:|:-----:|:-----:|:-----:|:-----:|
> |  CIFAR | CGL | 6.04  |  3.21   | 0.22 | 5.07 | 2.20 | 0.18 |  6.12 | 1.97| 0.19 | 7.95 | 1.59 | 0.25 | 10.75 | 1.51  |  0.33   |
> |    -10     | Ours    | **5.68**   | **2.32** | **0.19** | **4.40**|  **2.06** | **0.16** | **5.89** | **1.90** | **0.19** | **7.64** | **1.56** | **0.23** | **9.80** | **0.99** | **0.29** |
> | CIFAR   | CGL |25.55 | **1.85** | 0.90 |  22.75 | **3.11** |  **0.80** | 26.52 | 1.66 | 0.91 |  26.81  | 3.49  | 0.95  | 31.37  |  1.45  |  1.09 |
> |   -100    | Ours     | **25.52** | 3.81 | **0.90** | **21.44** | 7.22 |  0.82 | **26.29** | **1.66** | **0.91** | **26.66** | **2.59** | **0.94** |  **31.14** | **1.43** | **1.09** |
>
> * In the peer-based approach, our proposed method outperforms PCL and L-MCL in all the measurements to evaluate student generalization. PCL, in particular, requires temporal mean networks (TMN) as well as multiple peers and employs TMNs in proportion to the number of peers. As a result, PCL requires nearly double the number of parameters in training time to improve accuracy. Furthermore, unlike MCL, our method does not require any complex samplers, whereas MCL requires it to design self-supervised contrastive loss. Hence, our method is superior to PCL and L-MCL in generalizing a student model and even regarding training efficiency.
> * In the network-based comparison, our proposed method consistently outperforms CGL on CIFAR-10 in three measurements. However, on CIFAR-100, CGL also shows good calibration performance for the ResNet family, but in terms of ERR, which is the most important, we would like to address that our method is still consistently superior.
>
> The most recent manuscript includes PCL and CGL experimental results. If our paper is accepted, we will include all results, summarizing a discussion of L-MCL in the camera-ready version.
>
> We greatly appreciate the time and effort you spent reviewing our paper, which we are certain will make our work better.

---

### Official Review · Reviewer_D2AU · 2022-10-26

**Confidence:** 4
**Correctness:** 4
**Technical Novelty And Significance:** 2
**Empirical Novelty And Significance:** 3
**Recommendation:** 6

**Clarity, Quality, Novelty And Reproducibility:**

The paper is well-written and clear. As mentioned earlier in the review, while some / most of the component ideas such as label shift, PC-softmax, and the principle of enriching diversity in an ensemble for distillation have been observed before - the framework to put these together for online distillation is novel.
The experimental section is well-written and provides enough details for reproducibility.


**Strength And Weaknesses:**

Strengths:
+ The paper is written and structured well.
+ While many of the component ideas have been introduced before, incorporating such specialists in a complete framework of online distillation is novel and interesting.
+ The empirical analysis is thorough.

Weaknesses:
- The paper has a number of typos that are sometimes distracting. e.g.
  - On page 3: “... both categories, using the extra modules as ever.”
  - On page 4: “... target distribution while only with samples …”
  - On page 18: “... optimal exposure differs from models…”


**Summary Of The Paper:**

The paper proposes a framework for online knowledge distillation from an intentionally diversified ensemble of teachers. A common hypothesis for the success of distillation from an ensemble is that diverse teacher models increase the efficacy of the ensemble. The paper provides a formal framework to enhance teacher diversity by introducing a prior shift - effectively turning each teacher head into a “specialist”. The paper includes a robust empirical analysis and demonstrates that the proposed method significantly improves student performance with respect to both accuracy and calibration.

**Summary Of The Review:**

The paper proposes a new framework for online distillation with the intent to increase diversity among the teacher heads and demonstrates significant improvements over prior approaches in benchmark datasets.

---

> ### Author Response · Authors · 2022-11-11
> **Response to Reviewer D2AU**
>
> We would like to thank you for your valuable and detailed comments.
>
> Based on your suggestions, we have clarified our writing in the manuscript as follows:
>
> 1) On page 3: “… both categories, using the extra modules as ever.”$\rightarrow$“Chen et al. (2020); Kim et al.(2021) can be applied to both network- and peer-based approaches, but they require extra modules as well.”
>
> 2) On page 4: “...target distribution while only with samples...”$\rightarrow$“...target distribution only with the samples generated from a distribution we have.”
>
> 3) On page 18: “...optimal exposure differs from models...”$\rightarrow$“We choose different exposure for models as they differ in deep network architecture and the ratio of peer/shared parameters.”
>
> We greatly appreciate the time and effort you spent reviewing our paper, which we are certain will make our work better.

---

### Official Review · Reviewer_uYCq · 2022-12-11

**Confidence:** 3
**Correctness:** 3
**Technical Novelty And Significance:** 2
**Empirical Novelty And Significance:** 2
**Recommendation:** 3

**Clarity, Quality, Novelty And Reproducibility:**

The paper is mostly clear and the method is explained well so it should be reproducible. The novelty is somewhat limited as methods from previous works are combined without a clear explanation why using them is the best option.

Some specific questions:
- What is the motivation for the label prior shift for training models in an ensemble? MoEs are mentioned, but I missed an explanation if that particular technique was employed there and what its success was.
- “notably expected calibration error” – why is this finding notable?


**Strength And Weaknesses:**

Strenghs:
- The paper is mostly clear.
- The Figure 1 is a very nice summary of the work.

Weaknesses:
- One limitation is that the setup seems very specialized (on-line distillation, peer networks). I don’t see why the technique is specific for that setup - diversifying the teachers (which is the main idea behind the authors’ method) could be well applied to off-line distillation or non-peer networks.
- The baselines are not explained. It is not clear why these are chosen over other options (there are a lot of distillation methods one could compare to).
- The ablation section is very useful to have, but I would like to see ablations for why using each of the steps suggested by the authors’ is useful. For example, does PC-Softmax  help? Is importance sampling actually good compared to sampling the data instead? (checking this even on a small dataset would be useful) What about the effect of the peer network choice in the light of your method, as opposed to using completely separate networks?
- When averaging the teachers, why not weigh them differently depending on what labels they were trained on? Since each teacher focuses on a different set of labels, they should be experts with respect to such labels. It seems that averaging the teacher's predictions is naive and ignores useful information about the teachers.
- As an application of label prior shift for diversifying the teachers, Why not try it in non-on-line distillation? Is there anything special about on-line distillation that diversifying the teachers is particularly interesting compared to off-line distillation?


**Summary Of The Paper:**

The authors propose to diversify the models in an ensemble forming the teachers in distillation. The specific setup the authors consider is  on-line distillation (where the teachers are trained in parallel with the student) and peer-based (where the models share parts of their weights).

The proposed approach to achieve the aforementioned diversification of the teachers is based on varying the prior distributions of the labels in the train data used for training each teacher. Then, as an efficient approach to this, importance sampling is employed to avoid sampling from the data separately for each teacher. Then, before averaging the teacher outputs for the student supervision, the "Post-Compensated Softmax" loss correction is employed from a previous work.

The comparison against selected baselines shows improvements. Ablations are provided where the authors measure the effect of the teacher's diversity and the number of teachers within the proposed framework.

**Summary Of The Review:**

The use of the methods described by the authors looks interesting and promising. I suggest conducting more ablations and adding explanations for the decisions to strengthen the paper before resubmission.

---

> ### Author Response · Authors · 2022-12-13
> **Response to Reviewer uYCq**
>
> > One limitation is that the setup seems very specialized (on-line distillation, peer networks). I don’t see why the technique is specific for that setup - diversifying the teachers (which is the main idea behind the authors’ method) could be well applied to off-line distillation or non-peer networks.
>
> > As an application of label prior shift for diversifying the teachers, Why not try it in non-on-line distillation? Is there anything special about on-line distillation that diversifying the teachers is particularly interesting compared to off-line distillation?
>
> Thanks for your suggestion to use our proposed method for offline distillation. It seems interesting and applicable to offline distillation in a network-based architecture. If our paper is accepted, we will include the results in Appendix. In terms of distinguishing performance for online KD from offline distillation, Tian et al. [1] show in Table 3 that one of the online KD methods, “ONE,” which leverages an ensemble of multiple teachers, leads to almost better accuracy than solely using various offline methods. We would like to address the fact that our online KD outperforms ONE by diversifying the teachers, as shown in Table.1 and Table.2.
> Regarding the applicability of our method for non-peer networks, we show that our method can be applied to non-peer networks through Appendix F.1.
>
> > The baselines are not explained. It is not clear why these are chosen over other options (there are a lot of distillation methods one could compare to).
>
> We chose baselines in peer-based online KD works for Section 4, and network-based online KD works for Appendix F.1 for the additional case study. Although we briefly described each method in Related Work, this seems more required to describe the criteria. If our paper is accepted, we will thoroughly introduce the selected methods and compare them to our design methodology in the Appendix.
>
> > What is the motivation for the label prior shift for training models in an ensemble? MoEs are mentioned, but I missed an explanation if that particular technique was employed there and what its success was.
>
> We intended to solve peer homogenization in a distinctive and more straightforward approach than the previous works. The ensemble can be strong when many different peers are seemingly various but not extremely biased. We claimed that MoEs are an example of a helpful approach in that we can train each expert in a separate domain.
>
> > The ablation section is very useful to have, but I would like to see ablations for why using each of the steps suggested by the authors’ is useful. For example, does PC-Softmax help? Is importance sampling actually good compared to sampling the data instead? (checking this even on a small dataset would be useful) What about the effect of the peer network choice in the light of your method, as opposed to using completely separate networks?
>
> * Appendix C.1 and Appendix F.3 show that PC-Softmax is a better option than Softmax in that it corrects peers’ likelihoods that are elaborately different from the original distribution to facilitate the model averaging. As shown in Figure 4, a probability-based ensemble is more useful than a logit-based ensemble in forming an accurate and calibrated ensemble prediction. It also demonstrates that when the peers are specialists, PC-Softmax is more useful than PC-logit in the calibration.
> * Thanks for your suggestion to use sampling from imbalanced datasets. This is a good idea. We will test it.
> * Our proposed method is suitable for both peer-based and network-based architectures and exhibits outstanding calibration error.
>
> > When averaging the teachers, why not weigh them differently depending on what labels they were trained on? Since each teacher focuses on a different set of labels, they should be experts with respect to such labels. It seems that averaging the teacher's predictions is naive and ignores useful information about the teachers.
>
> In this paper, we want to address that our teacher has not to be trained on a separate label set. Instead, we want to generate diverse classifiers that resort to different features by varying the training data distribution. Thus, this paper informs that the model averaging, similar to Bagging, would suffice to combine the posteriors of different feature representations obtained via PC-Softmax on the various teacher classifiers. We appreciate your advice and suggestion to weigh teachers before they are combined. We will test the idea.
>
> > “notably expected calibration error” – why is this finding notable?
>
> Table.1 shows that our method is remarkably better performance in the ECE than the comparison in terms of error rate.
>
> We greatly appreciate the time and effort you spent reviewing our paper, which we are certain will make our work better.
>
>
> * Reference
>
> [1] Yonglong Tian et al., Contrastive Representation Distillation, ICLR, 2020

---

### Decision · Program_Chairs · 2023-01-20

**Decision:**

Reject

**Justification For Why Not Higher Score:**

Limited technical depth -- while a simple and intuitive idea, the key components are fairly standard in studies on label shift. The novelty appears to be in their application to the distillation setting, for which a more detailed discussion of why this is preferable to other ensemble generation strategies would be useful.

**Justification For Why Not Lower Score:**

N/A

**Metareview: Summary, Strengths And Weaknesses:**

The paper proposes a distillation strategy that leverages an ensemble of multiple teachers. These teachers are trained with different marginal label distributions. The resulting predictions are thus adjusted prior to ensembling, and the resulting average prediction is fed to a student model.

Reviewers were somewhat mixed on the paper, with concerns mostly on the empirical results and novelty. The response featured some new results, following which reviewer scores leaned more positive. The opinion of an additional domain expert was sought, which was a little more negative owing to the novelty and technical depth concerns. In my reading, I concur with these. The basic idea is interesting, and the paper is generally well written. The technical depth is a bit limited, with Section 3 essentially formalizing the strategy mentioned above. The ideas of importance sampling to construct the loss, and adjusting the logits to account for a label shift, are natural and fairly standard. It is also not immediately clear from the exposition what advantages the proposed scheme have over other means of constructing an ensemble of teachers; certainly the proposal generalizes training models on different subsets of classes, but are there deeper reasons to favour the proposal over, say, input clusters?

Overall this is certainly a decent paper, but is just a bit below the bar at present.

Minor comments:
- The idea of constructing specialist models in distillation is also explored in Section 4 of Hinton et al., 2015.
- Theorem 1 is an elementary consequence of the importance weighting identity. I would just state the result rather than wrap it in a theorem. Certainly Theorem 1 is overkill, and potentially a bit misleading unless one clearly states this is a standard result.
- Equation 2, use \left[ and \right].
- "manually adjusting logit values (Ren et al., 2020)" -> the cited paper changes the loss function. It does not appear to adjust logit values as stated.
- Appendix C.3 seems to present a new theoretical result that is not referenced in the body. This is not advisable.


**Summary Of Ac-Reviewer Meeting:**

N/A